# NFAT2 is a critical regulator of the anergic phenotype in chronic lymphocytic leukaemia

Melanie Märklin [1], Jonas S. Heitmann[1], Alexander R. Fuchs[1], Felicia M. Truckenmüller[1], Michael Gutknecht[1], Stefanie Bugl[1], Sebastian J. Saur[1], Juliane Lazarus[1], Ursula Kohlhofer[2], Leticia Quintanilla-Martinez [2], Hans-Georg Rammensee[3], Helmut R. Salih[1,4], Hans-Georg Kopp[1], Michael Haap[5], Andreas Kirschniak[6], Lothar Kanz[1], Anjana Rao[7], Stefan Wirths[1] & Martin R. Müller[1]

Chronic lymphocytic leukaemia (CLL) is a clonal disorder of mature B cells. Most patients are characterised by an indolent disease course and an anergic phenotype of their leukaemia cells, which refers to a state of unresponsiveness to B cell receptor stimulation. Up to 10% of CLL patients transform from an indolent subtype to an aggressive form of B cell lymphoma over time (Richter´s syndrome) and show a significantly worse treatment outcome. Here we show that B cell-specific ablation of *Nfat2* leads to the loss of the anergic phenotype culminating in a significantly compromised life expectancy and transformation to aggressive disease. We further define a gene expression signature of anergic CLL cells consisting of several NFAT2-dependent genes including *Cbl-b*, *Grail*, *Egr2* and *Lck*. In summary, this study identifies NFAT2 as a crucial regulator of the anergic phenotype in CLL.

[1] Department of Oncology, Haematology and Immunology, University of Tübingen, Tübingen 72076, Germany. [2] Department of Pathology, University of Tübingen, Tübingen 72076, Germany. [3] Department of Immunology, University of Tübingen, Tübingen 72076, Germany. [4] Clinical Collaboration Unit Translational Immunology, German Cancer Consortium (DKTK) and German Cancer Research Center (DKFZ), Heidelberg 69120, Germany. [5] Department of Endocrinology, Diabetology, Clinical Pathology and Metabolism, University of Tübingen, Tübingen 72076, Germany. [6] Department of Surgery, University of Tübingen, Tübingen 72076, Germany. [7] La Jolla Institute of Allergy and Immunology, La Jolla, CA 92037, USA. Correspondence and requests for materials should be addressed to M.R.M. (email: martin.mueller@med.uni-tuebingen.de)

Chronic lymphocytic leukaemia (CLL) is a clonal disorder of mature B cells characterised by the expression of CD19, CD23 and CD5[1]. While treatment options have improved substantially through the introduction of monoclonal antibodies (rituximab, obinutuzumab and ofatumumab)[2, 3] and kinase inhibitors (ibrutinib and idelalisib)[4, 5], most patients will still succumb eventually to disease complications. With respect to prognosis, CLL constitutes a heterogeneous disorder with the majority of patients exhibiting an indolent course for many years and others progressing rapidly and requiring early treatment[6]. Up to 10% of CLL patients transform from an indolent subtype to an aggressive form of B cell lymphoma over time (Richter's syndrome) and show a significantly worse treatment outcome[7]. Unmutated immunoglobulin heavy chain (IGHV) status as well as high expression of ZAP70 and CD38[8] are more common in a subgroup of patients with enhanced responsiveness to stimulation of the BCR complex and more aggressive disease. Patients with a more indolent course are typically characterised by an anergic B cell phenotype referring to unresponsiveness to BCR stimulation and essential lack of phosphotyrosine induction and calcium flux[9–11]. Previous studies have demonstrated that anergic CLL cells exhibit constitutive activation of ERK1/2 and NFAT2[11] as well as low or absent expression of ZAP70, CD38 and surface IgM[9]. Anergic CLL cells have also been characterised to exhibit depression of PRDM1 (BLIMP-1) expression and a subsequently compromised differentiation capacity[12]. Furthermore, there is evidence that anergy can be a mechanism of cell survival and that it could represent a potential target for leukaemia treatment[6, 9].

Upon dephosphorylation by calcineurin, NFAT transcription factors translocate to the nucleus where they orchestrate developmental and activation programs in diverse cell types[13–15]. They were originally discovered as master regulators of T cell differentiation and activation[16, 17] but also have an important function in the induction of T cell anergy and exhaustion[18–20]. The family member NFAT2 has been demonstrated to be important for the generation of B1a cells[21, 22]. In the context of cancer, NFAT2 is overexpressed and constitutively activated in a subset of patients with human CLL[23, 24], and is involved in maintaining lymphoma cell survival and counteracting apoptosis in B cell lymphomas by regulating the expression of survival factors such as CD40L and BLYS[25, 26]. NFAT2 has also been shown to contribute to the development of resistance towards tyrosine kinase inhibitors in chronic myeloid leukaemia[27] and the regulation of MYC expression in pancreatic cancer[28].

The extensive involvement of NFAT transcription factors in the induction of T cell anergy[18, 20] and the overexpression of NFAT2 in CLL[23, 24] prompted us to investigate its precise function in this disease using the Eμ-TCL1 transgenic mouse model and primary human tissue samples from patients with CLL and Richter's syndrome. We show that B cell-specific deletion of Nfat2 induces the loss of the anergic phenotype and disease transformation to aggressive B cell lymphoma. Our study further demonstrates that NFAT2 regulates the expression of several anergy-associated genes (Cbl-b, Egr2, Grail and Lck) in CLL cells. In summary, we identify NFAT2 as a major anergy regulator in CLL responsible for the maintenance of an indolent disease phenotype.

## Results

**Loss of NFAT2 expression induces CLL acceleration.** As NFAT2 has been previously described to be significantly overexpressed and constitutively activated in CLL[23, 24, 29, 30], we analysed patients with indolent and aggressive forms of CLL for the expression of this transcription factor (Fig. 1a–c). The differentiation between indolent and aggressive disease was performed using clinical and prognostic parameters (time to first treatment, Binet stage, IGHV status, high-risk genetic aberrations) (Supplementary Table 1). We could detect a significant degree of NFAT2 overexpression in CLL cells as compared to physiological B cells on the mRNA as well as on the protein level which was substantially more pronounced in the indolent patient cohort (Fig. 1a–c). To analyse the role of NFAT2 in vivo, we generated TCL1 transgenic mice with a B cell-specific deletion of Nfat2. TCL1 transgenic mice develop a human-like CLL due to the expression of the TCL1 oncogene under the control of the Eμ enhancer. While TCL1 mice are widely regarded to be a suitable model for the aggressive subtype of human CLL due to their unmutated IGHV status[29, 30], TCL1 transgenic CLL cells also exhibit important features of anergic disease which is typically associated with an indolent clinical course. This is exemplified by compromised calcium mobilisation upon BCR engagement as well as reduced IgM surface expression (Fig. 1d and e).

To generate the experimental cohort of mice with TCL1 expression and Nfat2 ablation in the B cell compartment, Nfat2[fl/fl] Cd19-Cre mice with Cre recombinase expression under the control of the Cd19 promoter were bred with Eμ-TCL1 transgenic mice to yield Eμ-TCL1 Nfat2[fl/fl] Cd19-Cre (TCL1 Nfat2[−/−]) mice after two generations (Fig. 1f). Eμ-TCL1 Nfat2[fl/fl] (TCL1 Nfat2[+/+]) mice and Nfat2[fl/fl] (Nfat2[+/+]) mice served as controls. B cells from Eμ-TCL1 Nfat2[fl/fl] Cd19-Cre mice exhibited complete absence of NFAT2 expression as assessed by western blotting, while B cells from control animals showed intact expression of this protein (Fig. 1g and Supplementary Fig. 1a). Expression of NFAT1 on the other hand was found to be unaffected in B cells from conditional NFAT2 knockout animals (Supplementary Fig. 1b).

TCL1 transgenic mice with B cell-specific loss of Nfat2 showed a significant acceleration of leukaemia development (Fig. 2a, b). At 28 weeks of age, ~20% of peripheral blood cells exhibited the typical CD19[+]CD5[+] CLL phenotype in TCL1 Nfat2[+/+] mice while a mean value of 60% could be observed in TCL1 Nfat2[−/−] mice (Fig. 2b). Starting at around day 100, TCL1 Nfat2[−/−] mice exhibited a significant acceleration of leukaemia proliferation as compared to controls with intact NFAT2 expression as documented by significantly increased lymphocytes and CD19[+] CD5[+] CLL cells (Fig. 2c, d). An increase of the CD19[+]CD5[−] population in TCL1 Nfat2[−/−] mice during later disease stages was found to be caused by a significant downregulation of CD5 in the malignant cell population (Supplementary Fig. 2). Other lymphoid subpopulations remained unaffected by Nfat2 deletion (Supplementary Fig. 3). In vivo BrdU incorporation assays and Annexin V staining revealed a several fold increased proliferation and apoptosis rate of CD19[+]CD5[+] CLL cells from TCL1 Nfat2[−/−] mice as compared to TCL1 mice with intact NFAT2 expression demonstrating a highly accelerated tumour cell turnover (Fig. 2e). In line with these observations, a significantly higher percentage of CLL cells in the NFAT2-deficient cohort were in the G2/M and S phases of the cell cycle documenting increased aggressiveness of disease (Fig. 2f).

As a consequence of leukaemia acceleration, the mean life expectancy in the Nfat2-deleted cohort was profoundly reduced (201 vs. 325 days) (Fig. 3a). There was also a tendency towards shorter survival in animals with one intact and one deleted Nfat2 allele which did not reach statistical significance indicating a potential gene dosage effect (276 vs. 325 days) (Supplementary Fig. 4). Upon necropsy at an age of 36 weeks, NFAT2-deficient animals showed dramatically stronger adenopathy and splenomegaly than TCL1 transgenic mice with intact NFAT2 expression and wild-type controls (Fig. 3b and c). The spleen weight was approximately increased 10-fold in animals with Nfat2 deletion in their CLL cells as compared to animals with regular NFAT2

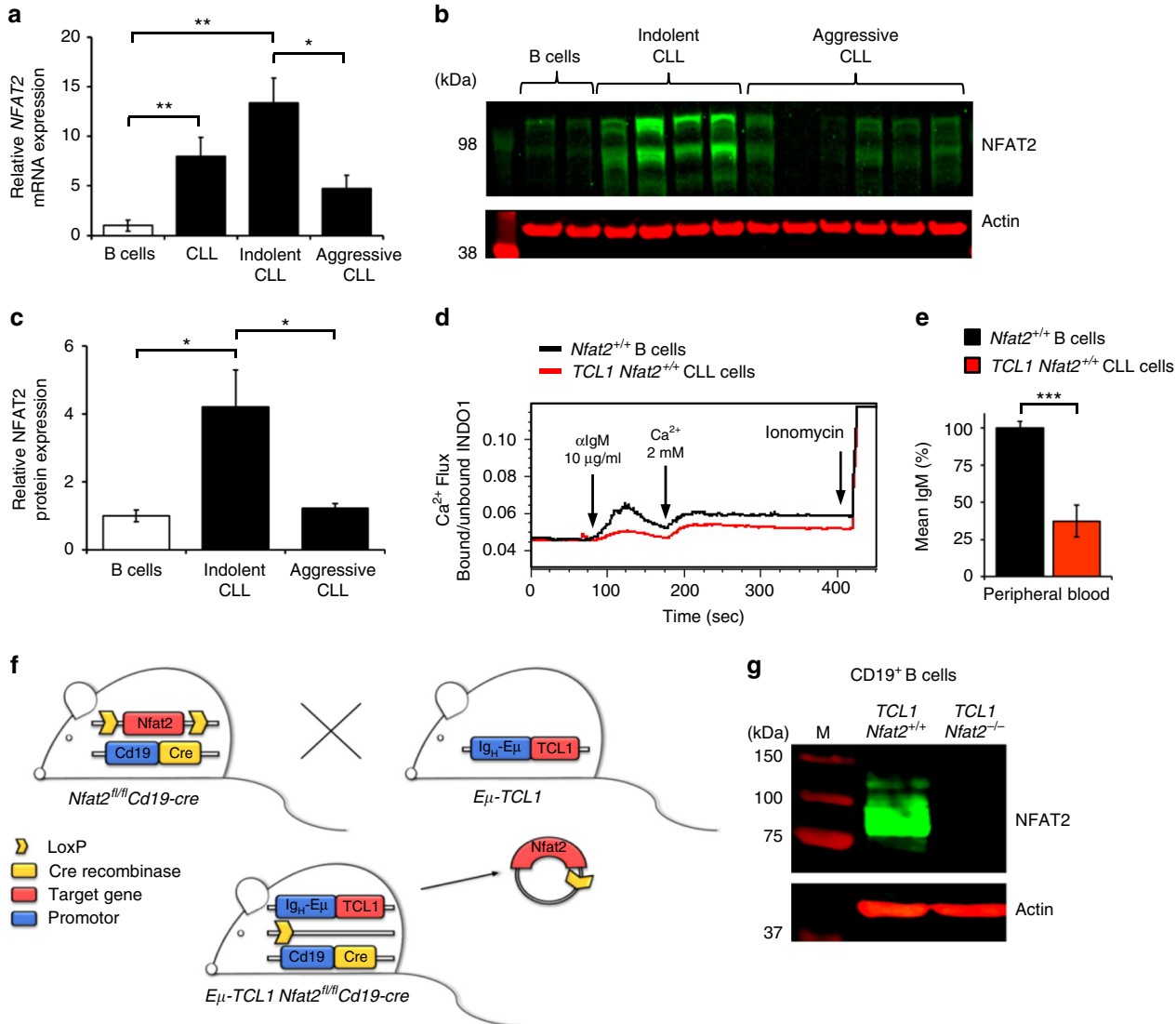

**Fig. 1** NFAT2 is overexpressed in CLL cells. **a** *NFAT2* mRNA expression in B cells from healthy volunteers ($n = 3$) and human CLL patients ($n = 12$), indolent CLL ($n = 6$) and aggressive CLL ($n = 6$) normalised to *Actin* assessed by RT-PCR (Welch's *t*-test, Mean ± S.E.M., *$P < 0.05$, **$P < 0.01$). **b** NFAT2 protein expression in physiological B cells ($n = 2$), indolent CLL ($n = 4$) and aggressive CLL ($n = 6$) samples assessed by western blotting. One representative blot is shown. **c** Quantitative analysis of NFAT2 protein expression in physiological B cells ($n = 10$), indolent CLL ($n = 7$) and aggressive CLL ($n = 8$) samples (Welch's *t*-test, mean ± S.E.M., *$P < 0.05$). **d** Ca$^{2+}$ flux of splenic CD19$^+$CD5$^+$ cells from 28-week-old *Nfat2$^{+/+}$* and *TCL1 Nfat2$^{+/+}$* mice after stimulation with 10 μg/ml αIgM. Ca$^{2+}$ was added after 3 min for extracellular flux. 1 μM ionomycin was added as a positive control. One representative experiment of three independent experiments is shown. **e** IgM surface expression on B cells from 28-week-old *Nfat2$^{+/+}$* and *TCL1 Nfat2$^{+/+}$* mice determined by flow cytometry ($n = 5$ per group) (Student's *t*-test, Mean ± S.E.M., *$P < 0.05$). **f** Mouse breeding scheme to generate *Eμ-TCL1 Nfat2$^{fl/fl}$ Cd19-Cre* mice. **g** NFAT2 protein expression in one representative CD19$^+$ splenic B cell sample from 28-week-old *TCL1 Nfat2$^{+/+}$* and *TCL1 Nfat2$^{-/-}$* mice assessed by western blotting

expression. This was accompanied by a higher degree of infiltration with malignant CD19$^+$CD5$^+$ CLL cells of the spleen, lymph nodes and bone marrow (Fig. 3d–g). Of note, the prevalence of CLL cells expressing the adverse prognostic markers ZAP70 and CD38 was significantly higher in the NFAT2-deficient cohort (Fig. 3h).

To verify that our observations were indeed CLL cell-specific phenomena, we subsequently performed transplantation experiments of TCL1 transgenic CLL cells with intact NFAT2 expression or with NFAT2 deficiency into immunocompromised NSG (NOD *scid* gamma) mice. As postulated, NSG mice transplanted with *TCL1 Nfat2$^{-/-}$* CLL cells exhibited significantly accelerated disease and a markedly reduced life expectancy when compared to NSG mice transplanted with TCL1 CLL cells with

intact NFAT2 expression (70 vs. 160 days) (Fig. 3i, j). Furthermore, cell proliferation was markedly increased and apoptosis reduced in *Nfat2*-deleted CLL cells (Supplementary Fig. 5). This clearly documents that the observed effects are CLL cell-intrinsic and not due to potential effects of *Nfat2* deletion in other B cell subsets.

**Nfat2 ablation leads to CLL transformation.** Up to 10% of human CLL patients transform from an indolent subtype to an aggressive form of B cell lymphoma during their disease course with a transformation rate of 0.5–1% per year (Richter's syndrome)[31]. These patients usually present with a rapid disease course and carry a dismal prognosis with a median survival of

approximately 1 year[32]. Loss of *CDKN2A*, disruption of *TP53*, activation of *MYC*, and mutations of *NOTCH1* have all been described to be associated with the development of Richter's transformation[33, 34]. The role of NFAT2 in this setting has not been studied.

To assess whether a transformation to aggressive lymphoma had occurred in our NFAT2-deficient TCL1 cohort, we performed an immunohistochemical analysis of spleens from mice at different stages of disease development (Fig. 4 and Supplementary Fig. 6). Mice with regular NFAT2 expression exhibited typical small lymphocytic infiltrates with clumped mature chromatin and low mitotic activity throughout the entire course of their disease (Fig. 4a, b and Supplementary Fig. 6) which are characteristic for CLL (small B cell lymphoma)[35].

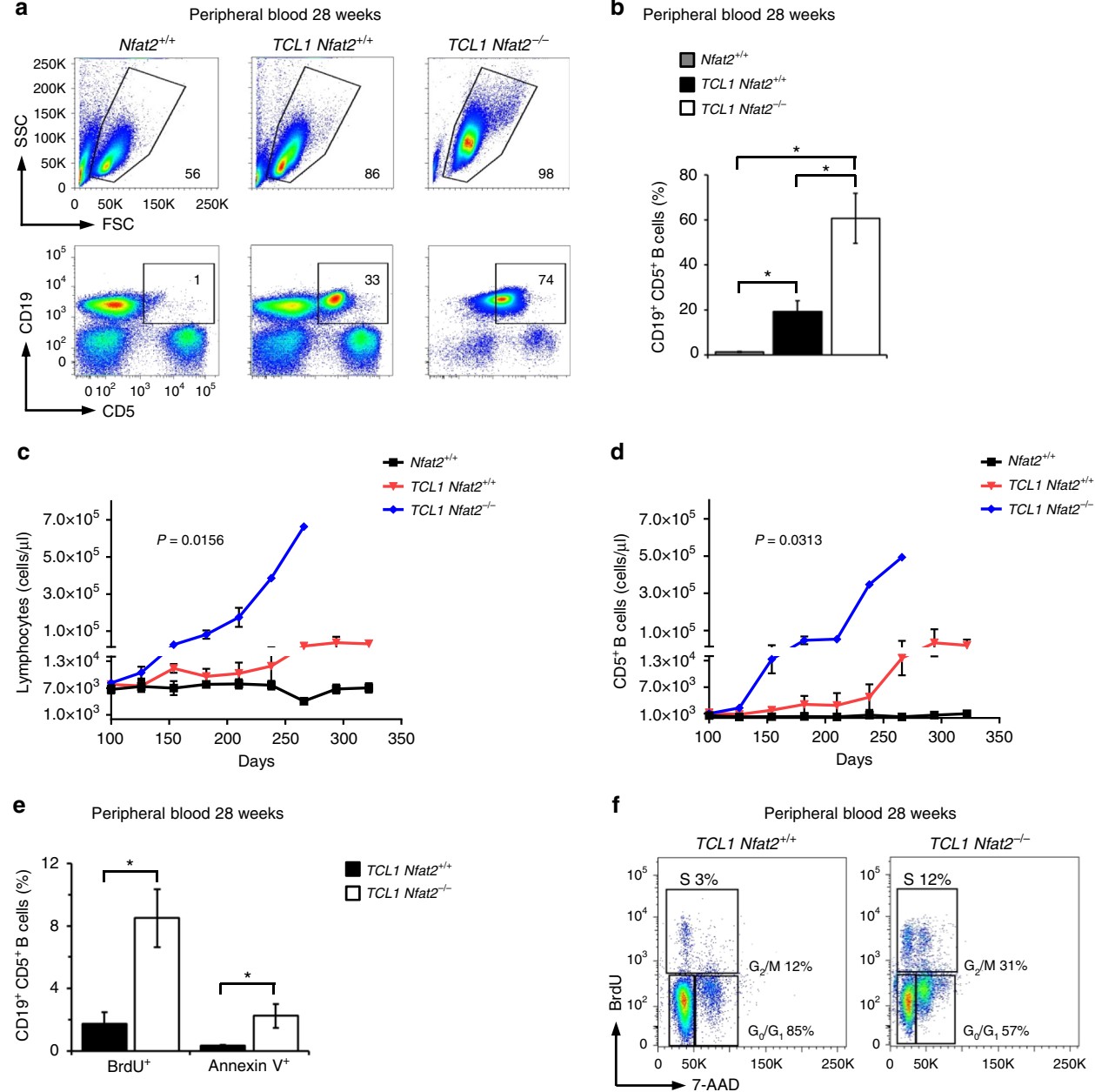

**Fig. 2** *Nfat2* deletion in *Eμ-TCL1* mice leads to significant acceleration of disease. **a** Flow cytometric analysis for CD19$^+$CD5$^+$ CLL cells of the peripheral blood of one representative mouse for the indicated genotypes at an age of 28 weeks. **b** Statistical analysis of the flow cytometry data from **a**. $n = 5$ animals per group were analysed (Welch's *t*-test, Mean ± S.E.M., *$P < 0.05$). **c** Absolute lymphocyte count of *Nfat2$^{+/+}$* ($n = 5$), *TCL1 Nfat2$^{+/+}$* ($n = 10$) and *TCL1 Nfat2$^{-/-}$* ($n = 10$) mice. Peripheral blood was harvested every 4 weeks and samples were analysed using an Advia 120 haematology analyser (Paired Wilcoxon test, Mean ± S.E.M.). **d** Expansion of CD19$^+$CD5$^+$ B cells in the peripheral blood of *Nfat2$^{+/+}$* ($n = 5$), *TCL1 Nfat2$^{+/+}$* ($n = 10$) and *TCL1 Nfat2$^{-/-}$* ($n = 10$) mice assessed by flow cytometry at the indicated time points (Paired Wilcoxon test, Mean ± S.E.M.). **e** Proliferation and apoptosis of CD19$^+$CD5$^+$ B cells in the peripheral blood of 28-week-old mice of the indicated genotypes ($n = 5$ per group). Mice were injected with 10 mM BrdU i.p. and peripheral blood cells were harvested after 48 h. CD19$^+$CD5$^+$ B cells were stained with BrdU and Annexin V antibodies and measured by flow cytometry ($n = 5$) (Welch's *t*-test, Mean ± S.E.M., *$P < 0.05$). **f** Cell cycle analysis of CD19$^+$CD5$^+$ B cells in the peripheral blood of 28-week-old mice of the indicated genotypes ($n = 5$ per group). Mice were injected with 10 mM BrdU i.p. and peripheral blood was harvested 24 h after injection. CD19$^+$CD5$^+$ B cells were stained with BrdU antibody and 7-AAD and subsequently analysed by flow cytometry

NFAT2-deficient mice on the other hand demonstrated clear evidence of transformation to an aggressive form of B cell lymphoma. Their spleens were diffusely infiltrated by medium to large cells with open blastic chromatin and a high mitotic index as documented by significantly increased Ki-67 expression. While the animals with intact NFAT2 expression showed a typical pattern of resident T cells in their spleens throughout disease development, only very few T cells could be detected in the organs of NFAT2-deficient mice at later stages of lymphoma development at 28 and 36 weeks of age (Fig. 4a, b and Supplementary Fig. 6). Furthermore, the lymphoid infiltrates in NFAT2-deficient mice only weakly expressed the B cell marker

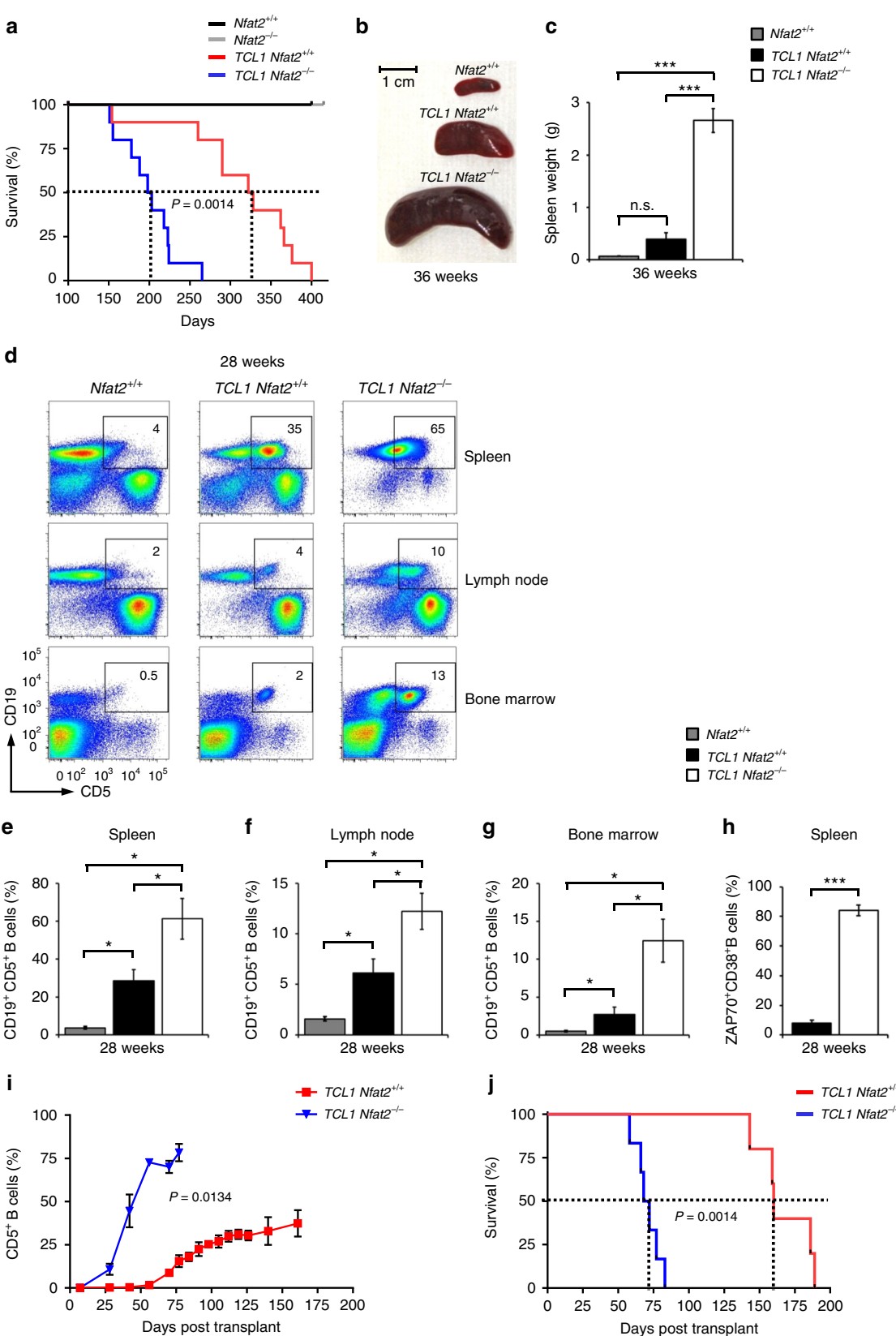

B220 but their B cell origin could be clearly documented by positivity for CD79a.

Taken together, these data unequivocally document that ablation of NFAT2 expression in the B cell compartment of TCL1 transgenic mice leads to transformation of CLL to an aggressive form of B cell lymphoma.

**NFAT2 regulates multiple anergy-associated genes.** To identify relevant NFAT2 target genes and to elucidate the mechanism by which the disease acceleration and transformation upon *Nfat2* ablation was mediated, we performed a gene expression analysis using Affymetrix microarrays. Here, we could detect a number of genes differentially regulated in the two leukaemia cohorts. The majority of these genes were either cell cycle regulators (Fig. 5a) or proteins involved in BCR signalling (Fig. 5b). Of note, we could also identify a significant downregulation of several genes that have been previously implicated in the regulation of the anergic state in physiological T and B cells in the *TCL1 Nfat2*[−/−] cohort. Among these were the E3 ligases *Cbl-b* and *Grail* as well as the transcription factor *Egr2* and the lymphocyte-specific protein kinase *Lck* (Fig. 5b). Cbl and Cbl-b have been demonstrated to possess an important role in anergy induction in B cells[36]. Other studies have elucidated that Cbl-b and Grail are significant regulators of $Ca^{2+}$-induced anergy in T cells[37, 38]. Egr2 has further been shown to be a critical transcription factor for the induction of T cell anergy in vitro and in vivo[39]. Lck has previously been demonstrated to be involved in BCR signaling[40] and has been shown to be overexpressed in CLL[41, 42]. It is well characterised for its important role in T cells, where it associates with the cytoplasmic tails of the co-receptors CD4 and CD8 to modulate signal transduction from the T cell receptor[43, 44] and possesses important functions in T cell activation and induction of T cell anergy[45].

Using microarrays we have identified the anergy-associated genes *Cbl-b*, *Grail*, *Egr2* and *Lck* to be differentially expressed in our two experimental cohorts with intact and ablated NFAT2 expression (Fig. 5b). Using polymerase chain rection with reverse transcription (RT-PCR) and western blotting, we were able to confirm a significantly reduced expression of this set of anergy-associated genes in NFAT2-deficient CLL cells both on the mRNA and protein level (Fig. 5c, d). To confirm that the identified anergy-associated genes were indeed target genes of a calcineurin-sensitive transcription factor, we subsequently demonstrated that their expression can be induced by BCR stimulation with αIgM and suppressed by treatment with the calcineurin inhibitors CsA and FK506 (Fig. 5e–h). We further show that the expression of *Lck* in BCR-stimulated TCL1-transgenic CLL cells can be efficiently downregulated using the NFAT peptide inhibitor VIVIT clearly identifying *Lck* as an NFAT target gene (Fig. 5h).

***Lck* is a direct target gene of NFAT2 in CLL.** Sequence analysis of the *Lck* gene locus has previously revealed a potential NFAT binding motif with the sequence TTTCATCAG in the *Lck* promotor (Fig. 6a)[46]. To biochemically define *Lck* as a direct target of NFAT2, we continued to perform chromatin immunoprecipitation (ChIP) in primary human CLL cells using a monoclonal antibody against NFAT2 and subsequent quantitative RT-PCR (Fig. 6b and Supplementary Table 3). Our results clearly demonstrate that NFAT2 binds to the *CD40L* locus, which is a well-defined target gene of NFAT2 in B cells and therefore served as a positive control, and also to the *Lck* locus (Fig. 6b). In summary, these results unequivocally define *Lck* as a direct target gene of NFAT2 in CLL cells.

To assess whether LCK was also overexpressed in human CLL, we subsequently analysed primary blood samples from patients with CLL while normal B cells from healthy donors served as controls (Fig. 6c–f). We demonstrate that LCK is substantially overexpressed and constitutively activated in leukaemia cells from patients with indolent CLL while its expression level and activation status in aggressive CLL was comparable to physiological B cells.

To decipher a potential role of LCK in BCR signalling in CLL cells, we went on to perform fluorescence co localisation assays (Fig. 6g, h and Supplementary Fig. 7). To demonstrate an interaction of LCK with the BCR complex-associated protein alpha chain CD79a, both proteins were labelled with monoclonal antibodies (Fig. 6g, panel 1) and subsequently incubated with oligonucleotide-coupled secondary antibodies (panel 2). In case of close proximity of both proteins, the oligonucleotides on the secondary antibodies are able to ligate (panel 3). A subsequent polymerisation and DNA amplification culminates in the development of a red fluorescent signal (panel 4). The Src kinase LYN which is a well characterised tyrosin kinase associated with the BCR complex served as a positive control. To assess the baseline levels of expression of LYN and LCK, single antibody stainings for the two proteins were also performed (Supplementary Fig. 7). In TCL1 CLL cells with intact NFAT2 expression, LCK clearly co localised with the BCR complex upon IgM stimulation while it was completely absent from the BCR under resting conditions (Fig. 6h, left panel). LYN on the other hand could be demonstrated to be associated with the BCR both under resting conditions and upon BCR engagement. In CLL cells from TCL1 mice deficient for NFAT2, LCK was completely absent from the BCR both at resting conditions and after BCR stimulation, while LYN co localised with the BCR upon αIgM treatment (Fig. 6h, right panel). In summary, these data show that LCK is a direct target gene of NFAT2 in CLL cells and co localises with CD79a upon BCR engagement.

**NFAT2 regulates the anergic phenotype in CLL.** We have shown that NFAT2 regulates the expression of several anergy-

---

**Fig. 3** *Nfat2* deletion in CLL cells leads to significantly compromised survival of *Eμ-TCL1* mice. **a** Kaplan–Meier survival plot of *TCL1 Nfat2*[+/+] mice ($n = 10$), *TCL1 Nfat2*[−/−] mice ($n = 10$) and *Nfat2*[+/+] ($n = 5$) and *Nfat2*[−/−] ($n = 5$) controls. Statistical significance was determined using the Log-rank (Mantel–Cox) test, $P = 0.0014$. **b** Spleen size of representative animals of the indicated genotypes at an age of 36 weeks. **c** Average spleen weight of $n = 5$ animals with the indicated genotypes at an age of 36 weeks (Welch's *t*-test, Mean ± S.E.M., ***$P < 0.005$, not significant (n.s.)). **d** Accumulation of CD19[+]CD5[+] B cells in the spleen, lymph nodes and bone marrow of one representative *Nfat2*[+/+], *TCL1 Nfat2*[+/+] and *TCL1 Nfat2*[−/−] mouse at an age of 28 weeks assessed by flow cytometry. **e–g** Accumulation of CD19[+]CD5[+] B cells in the spleen, lymph nodes and bone marrow of *TCL1 Nfat2*[+/+] mice, *TCL1 Nfat2*[−/−] and *Nfat2*[+/+] mice ($n = 5$ per group) assessed by flow cytometry at an age of 28 weeks (Welch's *t*-test, mean ± S.E.M., *$P < 0.05$). **h** ZAP70 and CD38 expression on CD19[+]CD5[+] B cells from animals with the indicated genotypes at an age of 28 weeks. $n = 5$ animals per group were analysed by flow cytometry (Welch's *t*-test, Mean ± S.E.M., ***$P < 0.005$). **i** Expansion of CD19[+]CD5[+] CLL cells in the peripheral blood of NSG mice transplanted with CLL cells from *TCL1 Nfat2*[+/+] ($n = 5$) or *TCL1 Nfat2*[−/−] mice ($n = 6$) assessed by flow cytometry at the indicated time points (Student's *t*-test, mean ± S.E.M., $p = 0,0134$). **j** Kaplan–Meier survival plot of NSG mice transplanted with CLL cells of the indicated genotype. Statistical significance was determined using the Log-rank (Mantel–Cox) test, $P = 0.0014$.

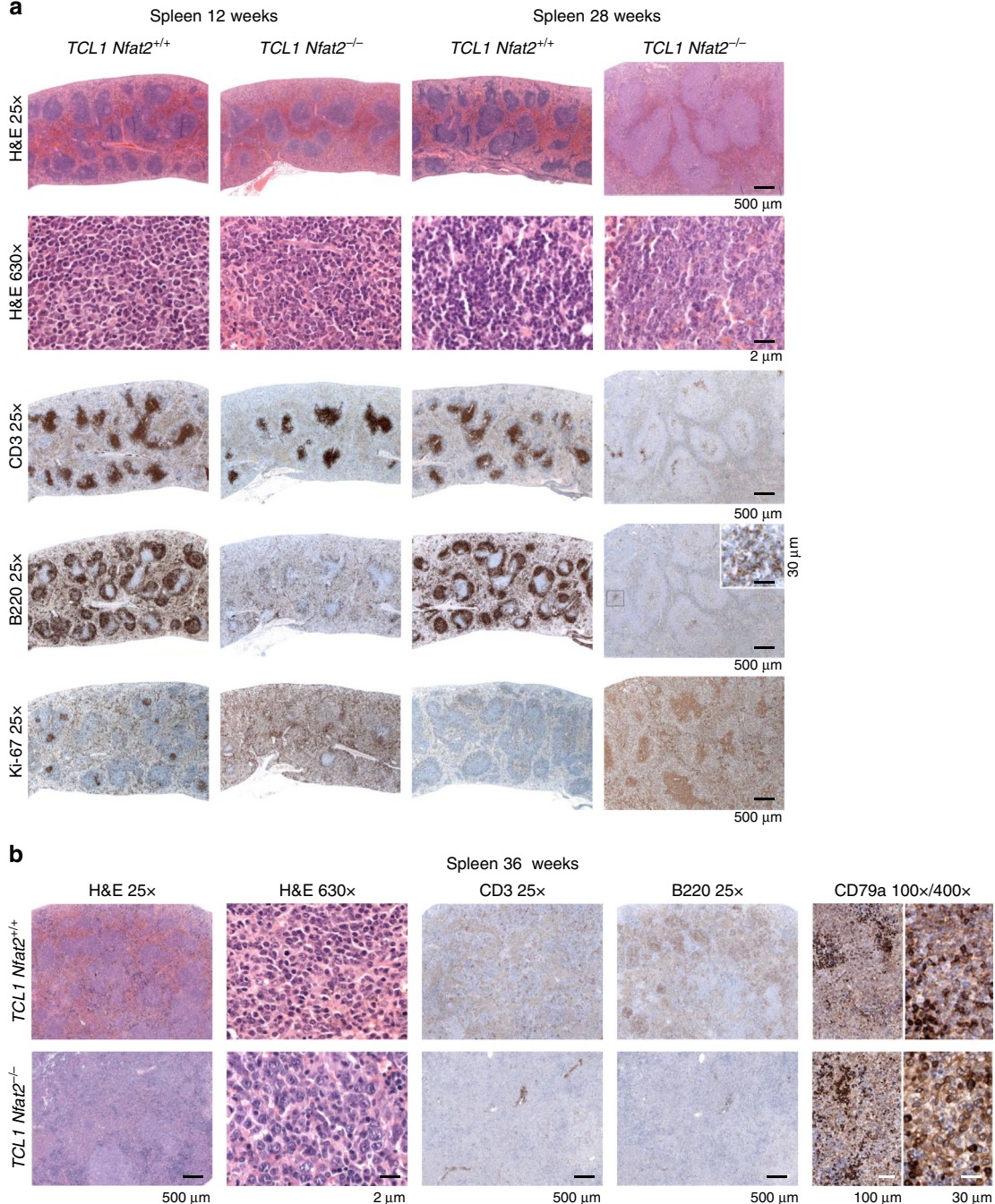

**Fig. 4** *Nfat2* ablation leads to histologic transformation of CLL to aggressive disease. **a** H&E staining of paraffin-embedded spleen sections of one representative *TCL1 Nfat2*$^{+/+}$ and *TCL1 Nfat2*$^{-/-}$ mouse at 12 and 28 weeks of age (*upper panels*) and immunohistochemical staining for CD3, B220 and Ki-67 (*lower panels*). For higher magnification of B220 and Ki-67 staining and CD79a staining in *TCL1 Nfat2*$^{-/-}$ mice and Staining of *Nfat2*$^{+/+}$control mice see also Supplementary Fig. 6. **b** H&E staining of paraffin-embedded spleen sections of representative *TCL1 Nfat2*$^{+/+}$ and *TCL1 Nfat2*$^{-/-}$ mice and immunohistochemical staining for CD3, B220 and CD79a at an age of 36 weeks

associated genes in CLL cells and that its loss leads to disease acceleration and histological transformation to aggressive lymphoma (Fig 4 and 5). To assess whether the leukaemic cells of NFAT2-deficient mice indeed had lost their anergic phenotype, we subsequently performed calcium mobilisation assays (Fig. 7a). While CLL cells from TCL1 mice with intact NFAT2 expression showed significantly reduced $Ca^{2+}$ mobilisation typical for the anergic state, leukaemic cells from NFAT2-deficient mice exhibited a strongly inducible BCR response with enhanced $Ca^{2+}$ mobilisation indicating the loss of the anergic phenotype. In line

with this observation, anergic CLL cells exhibited a significant downregulation of surface IgM expression as compared to physiological B cells, while CLL cells with *Nfat2* deletion showed an upregulation (Fig. 7b). Furthermore, we were able to detect a significant derepression of *Prdm1* expression in NFAT2-deficient CLL cells (Fig. 7c).

Anergic human CLL is characterised by constitutive activation of the BCR signalling cascade[11] while aggressive forms of the disease typically display an enhanced response to external BCR stimuli with consecutive stimulation of downstream targets[6].

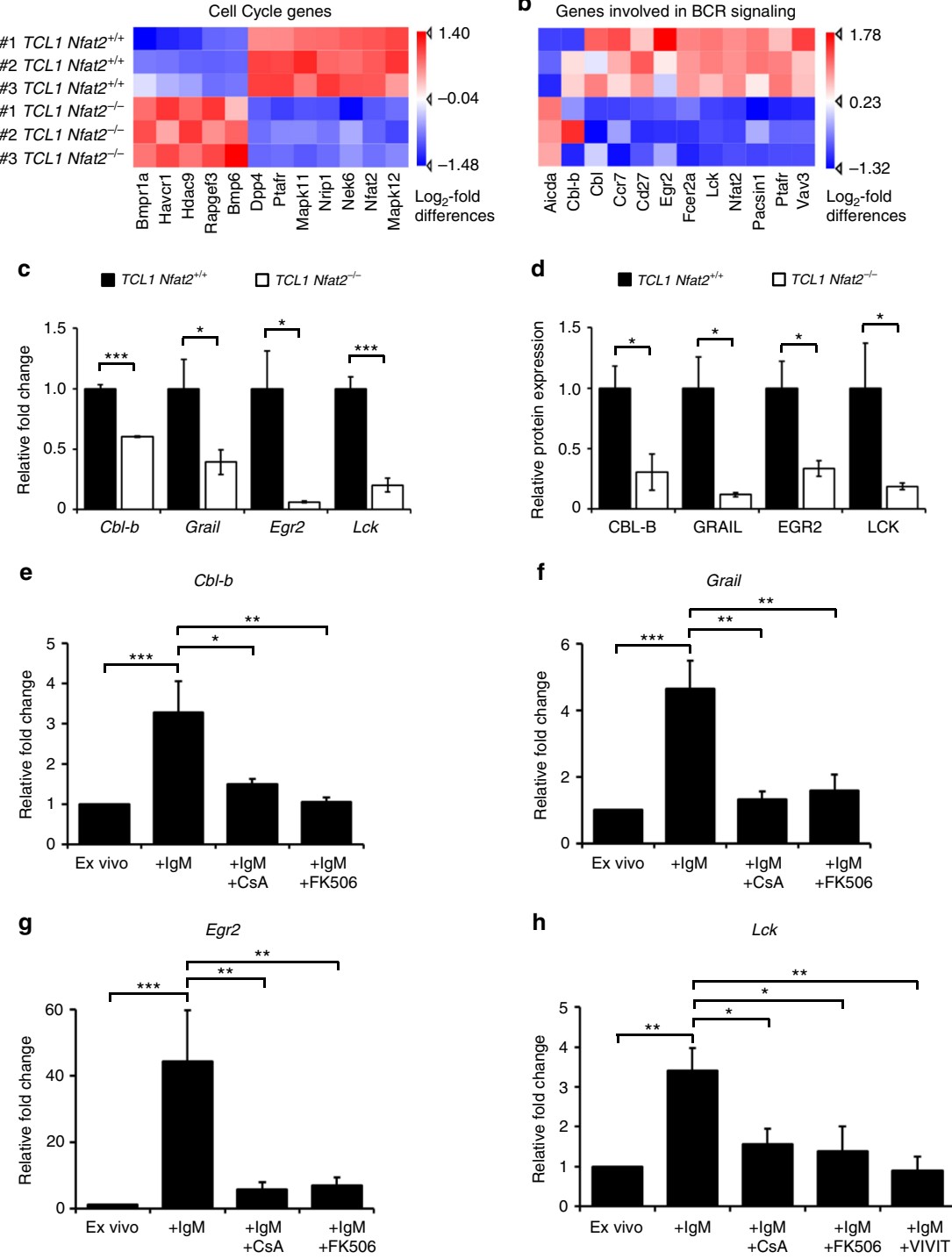

**Fig. 5** NFAT2 regulates the expression of multiple anergy-associated genes in CLL. **a**, **b** Microarray analysis of FACS-sorted splenic CD19+CD5+ CLL cells from 20-week-old *TCL1 Nfat2+/+* and *TCL1 Nfat2−/−* mice (*n* = 3 per group). Log2-fold differences of gene expressions are displayed in a hierarchical normalised heat map analysis which reveals changes in the expression of multiple cell cycle genes **a** and genes involved in BCR signalling **b**. **c** Relative gene expression of *Cbl-b*, *Grail*, *Egr2* and *Lck* mRNA normalised to *Actin* expression in ex vivo splenic CLL cells from 20-week-old *TCL1 Nfat2+/+* and *TCL1 Nfat2−/−* mice (*n* = 5 per group) assessed by qRT-PCR (Student's *t*-test, Mean ± S.E.M., \**P* < 0.05; \*\*\**P* < 0.005). **d** Quantification of protein expression of splenic CLL cells from 20-week-old *TCL1 Nfat2+/+* and *TCL1 Nfat2−/−* mice for CBL-B (*n* = 4), GRAIL (*n* = 3), EGR2 (*n* = 3) and LCK (*n* = 6) assessed by western blotting (Student's *t*-test, Mean ± S.E.M., \**P* < 0.05). **e**–**g** Splenic CLL cells from 20-week-old *TCL1 Nfat2+/+* mice were isolated and stimulated in vitro with 10 μg/ml αIgM, 10 μg/ml αIgM + 1 μM CsA or 10 μg/ml αIgM + 1 μM FK506 for 6 h. Relative mRNA expression of *Cbl-b* (*n* = 4) **f**, *Grail* (*n* = 3) **g** and *Egr2* (*n* = 5) **h** were assessed by qRT-PCR and normalised to *Actin* (One-way analysis of variance (ANOVA), Mean ± S.E.M., \**P* < 0.05, \*\**P* < 0.01, \*\*\**P* < 0.005). **h** Splenic CLL cells from 20-week-old *TCL1 Nfat2+/+* mice (*n* = 5) were isolated and stimulated in vitro with 10 μg/ml αIgM, 10 μg/ml αIgM + 1 μM CsA, 10 μg/ml αIgM + 1 μM FK506 or 10 μg/ml αIgM + 10 μM VIVIT peptide for 1 h. Relative *Lck* mRNA expression normalised to *Actin* was assessed by qRT-PCR (One-way ANOVA, mean ± S.E.M., \**P* < 0.05, \*\**P* < 0.01)

To test if this was also the case in our TCL1 transgenic NFAT2-deficient model, we went on to analyse the activation state of downstream signalling events by western blotting (Fig. 7d and Supplementary Fig. 8). Analysis of the BCR-associated kinase LYN[47] revealed no difference in total protein expression in *TCL1 Nfat2+/+* and *TCL1 Nfat2−/−* mice. The inactive form of LYN with an inhibitory phosphorylation of Tyr-507 however was clearly overexpressed in the *TCL1 Nfat2+/+* cohort demonstrating the anergic state of this cell population. SYK, which is normally

recruited to the BCR signalling complex upon external stimulation[48], was expressed at comparable levels in both cohorts. CLL cells of the two groups displayed no evidence of SYK phosphorylation *ex vivo*. While BCR stimulation clearly induced SYK phosphorylation and activation in the *Nfat2*-deleted cohort, anergic CLL cells showed no evidence of SYK phosphorylation after IgM stimulation. LCK was overexpressed and constitutively activated in the anergic CLL cells from *TCL1 Nfat2+/+* mice, while leukaemic cells from *TCL1 Nfat2−/−* mice showed no

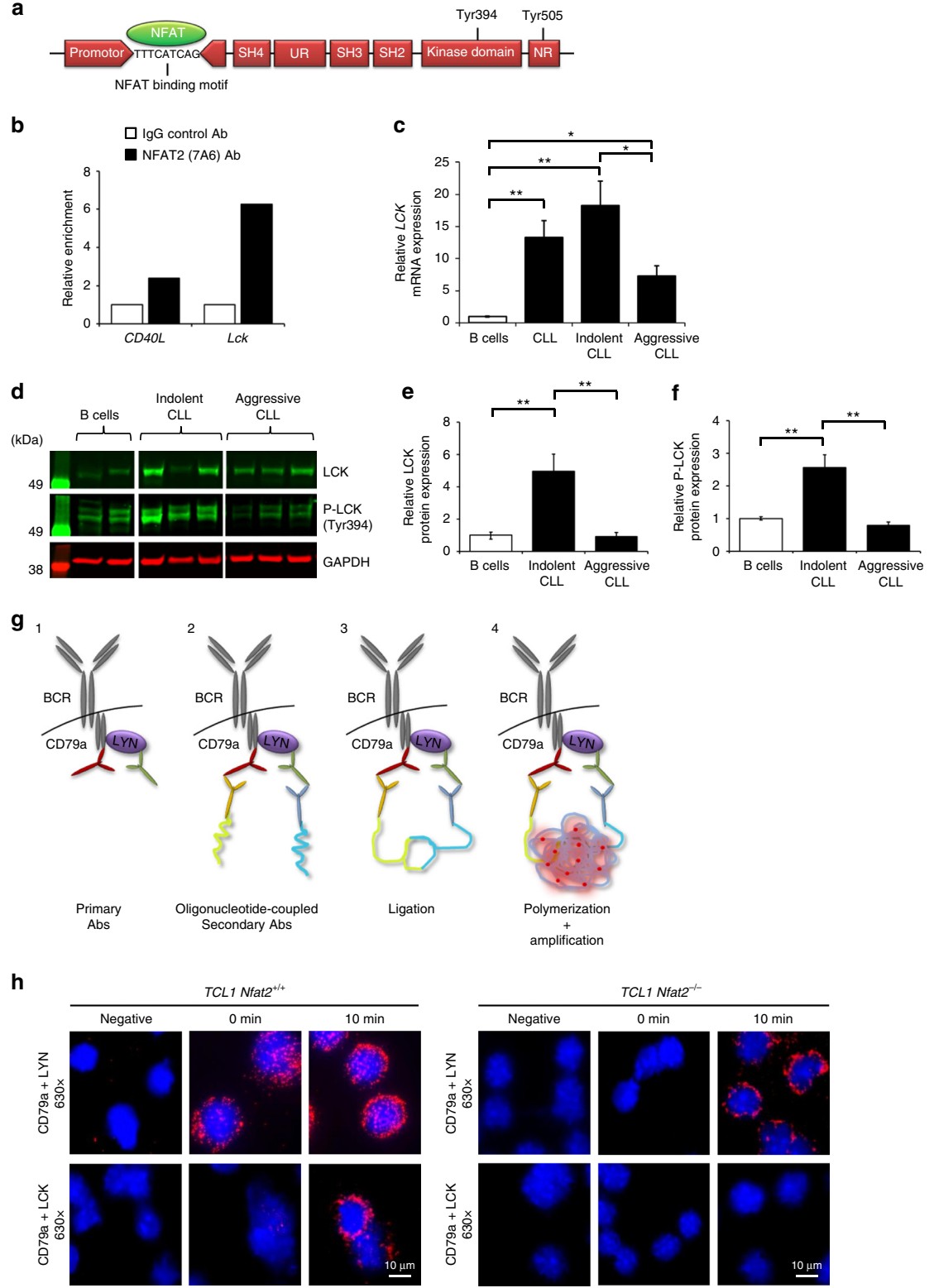

evidence of LCK activation upon αIgM treatment as assessed by the absence of the activating phosphorylation at Tyr-394. Expression of the inactive form of LCK with phosphorylation at Tyr-505 on the other hand was identical in the two cohorts. Anergic CLL cells from TCL1 mice with regular NFAT2 expression showed evidence of constitutive activation of ERK1/2 as it is the case in the human disease[11], whereas in CLL cells from TCL1 Nfat2[−/−] mice ERK1/2 was completely inactive without BCR stimulation and changed its activation status only after αIgM treatment. In summary, these data clearly document that Nfat2 ablation contributes to the loss of the anergic state in CLL.

To assess whether the NFAT2-LCK axis also possesses a function in human CLL, we determined NFAT2 and LCK expression levels in lymph node biopsies from patients with indolent CLL and Richter's syndrome (Fig. 7e–h and Supplementary Table 2). Interestingly, NFAT2 as well as LCK were clearly overexpressed in patients with indolent CLL and the typical anergic phenotype, whereas a substantial downregulation of these two proteins could be detected in patients with Richter's syndrome. Furthermore, we were able to demonstrate that CLL cells from patients with indolent disease typically exhibit compromised calcium mobilisation capacity indicating their anergic state (Supplementary Fig. 9). In line with these observations, we could detect a significant downregulation of Cdkn2A and Trp53, which are both known to be associated with Richter's syndrome in humans, in CLL cells from TCL1 Nfat2[−/−] mice (Supplementary Fig. 10). In summary, our data indicate that NFAT2 and LCK also possess a crucial function in the maintenance of the anergic phenotype in human CLL and in the pathogenesis of Richter's transformation.

## Discussion

In this study, we demonstrate that genetic ablation of the transcription factor NFAT2 leads to the loss of the anergic CLL phenotype and subsequent disease acceleration in vivo. We show that TCL1 transgenic mice with B cell-specific ablation of Nfat2 display a significantly shorter life expectancy (201 vs. 325 days) and transformation to aggressive B cell lymphoma. Using microarrays, we further define a molecular signature of anergy in CLL cells consisting of the E3 ligases Cbl-b and Grail, the transcription factor Egr2 and the lymphocyte-specific protein kinase Lck. In addition, we define Lck as a direct target gene of NFAT2 in CLL cells and demonstrate that while the NFAT2-LCK axis is constitutively activated in indolent CLL, it is inactive in tumour cells from patients with aggressive CLL or Richter's syndrome.

In anergic CLL, tonic stimulation of the BCR induces the recruitment of LCK and the activation of NFAT2, which

subsequently translocates to the nucleus to induce the transcription of several anergy-associated genes including Cbl-b, Grail, Egr2 as well as Lck (Fig. 8, left panel). In the absence of NFAT2, BCR stimulation leads to enhanced phosphotyrosine induction and calcium flux culminating in the activation of AKT and ERK and subsequent cell proliferation which can be observed in patients with aggressive forms of CLL or Richter's syndrome (Fig. 8, right panel).

A number of attempts to define the anergic phenotype of CLL have been undertaken in previous studies[9, 11, 23]. Constitutive phosphorylation of ERK1/2 in the absence of AKT activation, constitutive phosphorylation of MEK1/2 and NFAT2 overexpression were described to be associated with attenuated BCR signalling upon αIgM treatment in a subset of human CLL patients[11]. Another analysis observed NFAT2 overexpression and its constitutive activation in virtually all tested human CLL samples[23]. The authors of this study describe differences in the DNA binding capacity of NFAT2 upon IgM stimulation as the decisive factor. In yet another analysis, it was found that the anergic subset of CLL is characterised by constitutive ERK1/2 phosphorylation, low IgM surface expression and impairment of calcium mobilisation upon BCR engagement in vitro[9]. The authors of this study further demonstrate that anergic cells can be made susceptible to MAP kinase inhibitors by inhibiting NFAT signalling with VIVIT peptide and thus restoring BCR responsiveness. Another analysis could reveal that the anergic phenotype and a subsequent compromised differentiation capacity in CLL cells can be associated with transcriptional repression of Prdm1[12].

The results of our work are in line with the observations made by previous investigations. We detected overexpression of NFAT2 and LCK as well as constitutional activation of LCK in virtually all indolent CLL samples analysed. The TCL1 mouse model which was employed for the in vivo experiments exhibited all crucial features of anergic CLL cells like impairment of calcium mobilisation, downregulation of IgM and Prdm1 expression (Figs 1d, e and 7c) as well as constitutive activation of ERK1/2 (Fig. 7d). Abrogation of NFAT2 expression resulted in the loss of the anergic phenotype as documented by enhanced calcium mobilisation, upregulation of IgM and Prdm1 as well as an enhancement of the signalling response upon BCR engagement (Fig. 7a–d).

Animals with deletion of Nfat2 in their B cell compartment also exhibited a transformation of indolent CLL to aggressive B cell lymphoma (Fig. 4). Furthermore, our analysis documents that NFAT2 and LCK expression are significantly downregulated in patients with Richter's syndrome (Fig. 7e–h), a similar transformation which occurs in up to 10% of human CLL patients[31].

**Fig. 6** Lck is a direct target gene of NFAT2 and co localises with the BCR in CLL cells. **a** Schematic illustration of the Lck gene with a putative NFAT binding site. **b** Chromatin immunoprecipitation (ChIP) with CLL patient cells stimulated for 16 h with PMA/ionomycin. Cells were fixed with paraformaldehyde and DNA content was sheared by sonication. ChIP was performed with NFAT2 (7A6) and IgG control antibodies. The number of immunoprecipitated regions for each gene was calculated and normalised to the respective IgG control. One representative experiment of three independent experiments is shown. **c** LCK mRNA expression in untouched isolated physiological B cells from healthy volunteers ($n = 6$) and human CLL patients ($n = 11$) with indolent CLL ($n = 6$) and aggressive CLL ($n = 5$) normalised to GAPDH assessed by RT-PCR (Welch's t-test, mean ± S.E.M., *$P < 0.05$, **$P < 0.01$, ***$P < 0.001$). **d** Representative LCK protein expression and its activating phosphorylation at Tyr394 in physiological B cells ($n = 2$), indolent CLL ($n = 3$) and aggressive CLL ($n = 3$) cells assessed by western blotting. **e**, **f** Quantitative analysis of LCK **e** and P-LCK (Tyr394) **f** expression on western blots with physiological B cells ($n = 8$), indolent CLL ($n = 8$) and aggressive CLL ($n = 8$) cells (Welch's t-test, mean ± S.E.M., **$P < 0.01$). **g** Co localisation assay: Proteins were labelled with monoclonal antibodies from different species (1) and subsequently incubated with oligonucleotide-coupled secondary antibodies (2). In the case of close proximity of both proteins, the oligonucleotides on the secondary antibodies are able to ligate (3). A subsequent polymerisation and DNA amplification initiates the development of a red fluorescent signal (4), which can be detected in the fluorescence microscope. **h** Cytospins of PBMCs from 20-week-old TCL1 Nfat2[+/+] and TCL1 Nfat2[−/−] mice were either prepared without stimulation or after treatment with 20 μg/ml αIgM F(ab')$_2$ fragments for 10 min. Cells were stained with antibodies for CD79a and LCK. Co localisation of CD79a and LYN was used as a positive control. For negative controls, the primary antibody against CD79a was not added. DAPI was used for nuclear staining (630×)

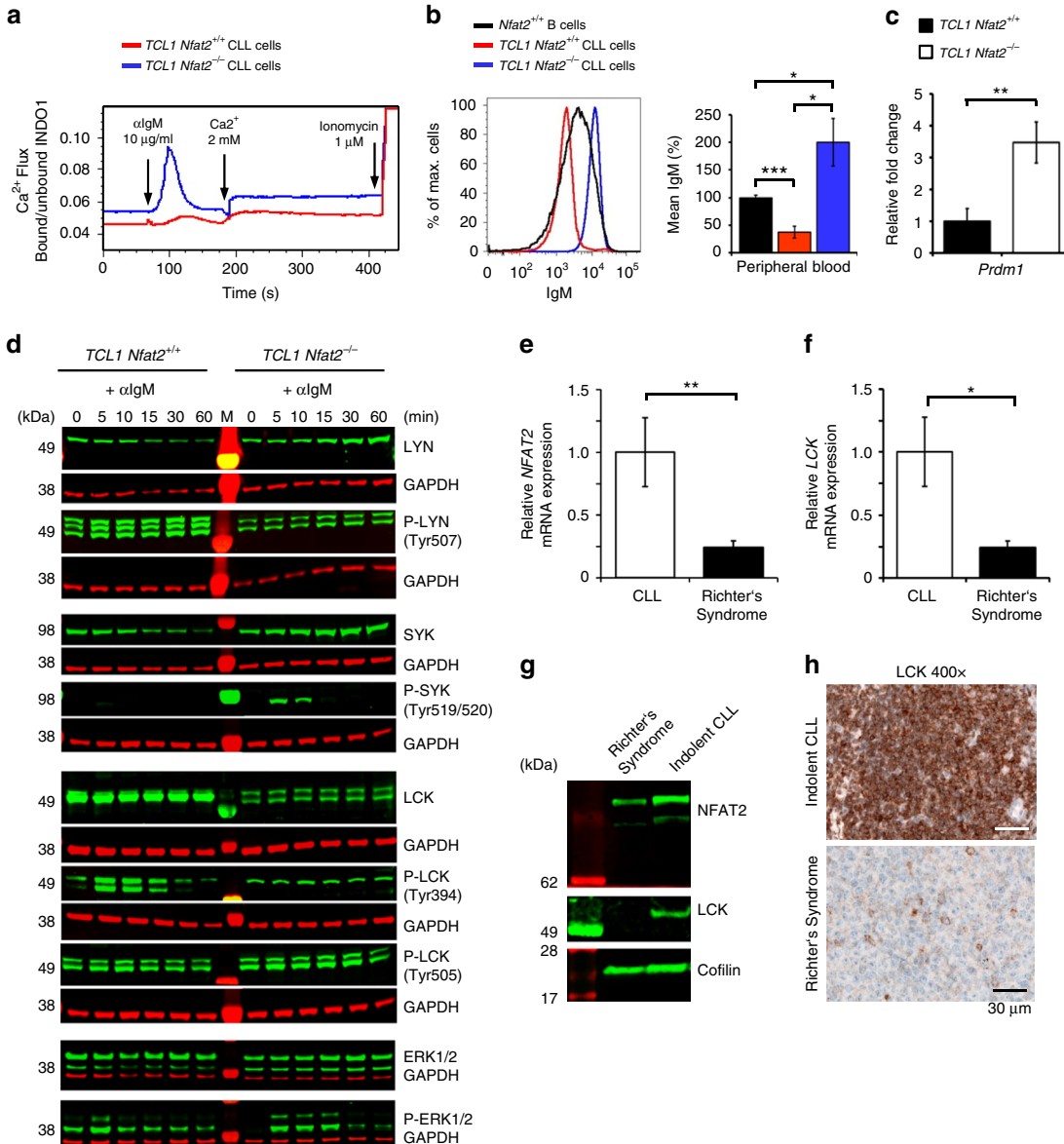

**Fig. 7** NFAT2 controls the anergic phenotype in CLL. **a** Ca$^{2+}$ flux of splenic CD19$^+$CD5$^+$ CLL cells from 28-week-old *TCL1 Nfat2*$^{+/+}$ and *TCL1 Nfat2*$^{-/-}$ mice after stimulation with 10 μg/ml αIgM. Ca$^{2+}$ was added after 3 min for extracellular flux. 1 μM ionomycin was added as a positive control. One representative result of three independent experiments is shown. **b** IgM surface expression on B cells from 28-week-old *Nfat2*$^{+/+}$, *TCL1 Nfat2*$^{+/+}$ and *TCL1 Nfat2*$^{-/-}$ mice determined by flow cytometry (*n* = 5 per group). A representative histogram plot (*left panel*) and a statistical analysis of all animals (*right panel*) are shown (Student's *t*-test, mean ± S.E.M., *P < 0.05, ***P < 0.005). **c** Relative *Prdm1* mRNA expression normalised to *Actin* in ex vivo splenic CLL cells from 28-week-old *TCL1 Nfat2*$^{+/+}$ and *TCL1 Nfat2*$^{-/-}$ mice (*n* = 6 per group) assessed by qRT-PCR (Student's *t*-test, mean ± S.E.M., **P < 0.01). **d** Splenic CD19$^+$CD5$^+$ CLL cells from 20-week-old *TCL1 Nfat2*$^{+/+}$ and *TCL1 Nfat2*$^{-/-}$ mice were stimulated in vitro with 20 μg/ml αIgM for the time points indicated. Total protein levels and phosphorylation status of LYN (Tyr507), SYK (Tyr519/520), LCK (Tyr394 and Tyr505) and ERK1/2 (Thr202/Tyr204 Thr185/Tyr187) were assessed by western blotting. One representative result of three independent experiments is shown. **e** Paraffin-embedded lymph node biopsies from human patients with CLL (*n* = 5) and Richter's syndrome (*n* = 9) were analysed for relative *NFAT2* mRNA expression normalised to *GAPDH* using qRT-PCR (Mann–Whitney test, Mean ± S.E.M., **P < 0.01). **f** Paraffin-embedded lymph node biopsies from human patients with CLL (*n* = 5) and Richter's syndrome (*n* = 9) were analysed for relative *LCK* mRNA expression normalised to *GAPDH* using qRT-PCR (Student's *t*-test, mean ± S.E.M., *P < 0.05). **g** NFAT2 and LCK protein expression in one representative patient with indolent CLL (*left*) and one patient with Richter's syndrome (*right*). **h** Immunohistochemical staining for LCK of a representative lymph node with indolent CLL and Richter's Syndrome (400×)

An important question to ask in this context is by which mechanism NFAT2 expression is lost during disease evolution and acceleration of CLL. While whole genome sequencing of human CLL samples provided no evidence for inactivating mutations in the regulatory region of the NFAT2 gene or of other components of the Calcineurin-NFAT signalling pathway, several lines of evidence point to epigenetic changes in the NFAT2 promotor region as a potential mode of downregulation of this

transcription factor[49]. Further studies in human tumour material are warranted to address this important question.

Our data clearly document that loss of NFAT2 expression is sufficient to induce lymphoma transformation in the TCL1 mouse model and suggest that NFAT2 is also a relevant factor in the pathogenesis of Richter's syndrome in human CLL patients. Our NFAT2-deficient TCL1 mouse model which resembles human Richter's syndrome in many important aspects might

therefore prove to be a helpful tool for the study of other aspects of this devastating disease. The results also suggest that pharmacological restoring of NFAT2-LCK signalling might be of therapeutic benefit in human patients with Richter's syndrome.

In summary, our data provide evidence that NFAT2 is a critical regulator of anergy in CLL and suggests that its inactivation can contribute to disease acceleration and transformation.

## Methods

**Mice.** Cd19-Cre mice were obtained from The Jackson Laboratory (stock #006785). Mice bearing a conditional allele of *Nfat2* (*Nfat2*$^{fl/fl}$) have been described previously[22]. The transgenic *Eμ-TCL1* mice were kindly provided by C. M. Croce (Columbus, OH)[47]. Mice were used on the C57BL/6 background (CRL 027) and maintained under specific pathogen-free conditions. Homozygous *TCL1 Nfat2*$^{fl/fl}$ *Cd19-Cre* (*TCL1 Nfat2*$^{-/-}$) mice with B cell-specific deletion of *Nfat2* were used as the experimental cohort, while *TCL1 Nfat2*$^{fl/fl}$ (*TCL1 Nfat2*$^{+/+}$) mice without *Nfat2* deletion served as controls. All mice were age and sex-matched and were killed at the indicated time points by overdosing inhalation anaesthesia. Mice exhibiting clinical signs of disease or >20% weight loss were removed early from the experimental cohort. Animal experiments were performed with the authorisation of the Institutional Animal Care and Use Committee of the University of Tübingen according to German federal and state regulations.

**CLL samples.** PBMCs from CLL patients were obtained by ficoll gradient centrifugation and CLL cells were isolated with a MACS-based B-CLL Cell Isolation Kit (Miltenyi, 130103466). Purity above 90% was confirmed by flow cytometry staining for CD19 (HIB19, eBioscience, 1:100). Physiological B cells from voluntary blood donors were isolated using the same procedure and were used as controls. Paraffin-embedded lymph node biopsies from patients with and without Richter's transformation were used for RNA isolation and qRT-PCR. All experiments were approved by the Ethics Committee of the University of Tübingen and written informed consent was obtained from all patients who contributed samples to this study.

**B cell isolation and stimulation.** Splenic B cells were negatively isolated with a MACS-based B Cell Isolation Kit (Miltenyi, 130090862) and purity above 90% was confirmed by flow cytometry staining for CD19 (eBio1D3, eBioscience, 1:100). B cells were stimulated *in vitro* with goat anti-mouse αIgM F(ab′)$_2$ fragments (Jackson ImmunoResearch) for the indicated times and with the respective concentrations. Cyclosporine A (CsA) (1 μM, Sigma Aldrich), FK506 (1 μM, Sigma Aldrich) and VIVIT peptide (10 μM, Tocris) were consistently added 1 h prior to αIgM stimulation.

**Peripheral blood analysis.** Retroorbital blood was collected and differential blood counts were obtained using an automated Bayer Advia 120 MultiSpecies Analyser

(Bayer HealthCare). For flow cytometric analysis (FACS-Canto II; BD Bioscience), red blood cells were lysed with ammonium chloride buffer (0.150 mM NH$_4$Cl, 0.1 mM EDTA, 0.150 mM KHCO$_3$) for 10 min on ice. CLL cells were stained with CD19-FITC (1:100), CD5-APC (1:80), B220-eFlour605 (1:40), IgM-PECy7 (1:200) antibodies (eBioscience).

**Transplantation experiments.** Total splenocytes from *TCL1 Nfat2*$^{-/-}$ and *TCL1 Nfat2*$^{+/+}$ mice (7–10 month old) were harvested and subsequently isolated by ficoll gradient centrifugation. Cells were stored in foetal calf seum (FCS) + 10% DMSO at −80 °C. After thawing, $2 \times 10^7$ cells were injected i.p. in 10 week old NOD.Cg-Prkdc$^{scid}$Il2rg$^{tm1Wjl}$/SzJ (NSG) mice. The peripheral blood was checked every 2 weeks for the expansion of CD19$^+$CD5$^+$ CLL cells by flow cytometry and survival of the mice was monitored.

**Flow cytometry.** Cell suspensions of different organs were lysed with ammonium chloride buffer (0.150 mM NH$_4$Cl, 0.1 mM EDTA, 0.150 mM KHCO$_3$) to eliminate erythrocytes followed by staining with CD19-FITC/APC (eBio1D3, 1:100), CD38-PE (clone 90, 1:100), IgM-PE/PECy7 (II/41, 1:200), IgD-eFlour450 (11-26c, 1:30), CD5-APC/Pacific Blue (53-7.3, 1:80), B220-eFlour605 (RA3-6B2, 1:40), CD3-FITC (145-2C11, 1:100), ZAP70-FITC (1E7.2, 1:20), CD21-FITC (eBio8D9, 1:100), CD23-PECy7 (B3B4, 1:160), CD93-PE (AA4.1, 1:80), LIN-Cocktail (CD11b-Biotin (M1/70, 1:130), CD3-Biotin (145-2C11, 1:130), Ter119-Biotin (Ter119, 1:130) and Gr1-Biotin (RB6-8C5, 1:130), Streptavidin-PerCP (1:160), Annexin V-Pacific Blue (1:25) and 7-AAD (1:100) (all from eBioscience). Cells were analysed with the Canto II cytometer (BD Bioscience). Data was obtained with the BD FACSDiva$^{TM}$ (BD) and analysed with the FlowJo 8.5.3 software. The sequential gating strategy for all FACS panels is provided in Supplementary Fig. 11.

**Proliferation assay.** BrdU solution (1 mM, 200 μl) was injected i.p. once every 24 h and mice were sacrificed and cell suspensions of different organs were prepared after 48 h. Cells were stained with CD19-FITC (eBio1D3, 1:100), IgM-PECy7 (II/41, 1:200), CD5-APC (53-7.3, 1:80), B220-eFlour605 (RA3-6B2, 1:40) (eBioscience) according to the BrdU Flow Kit APC Manual (BD, 552598) and analysed with the Canto II cytometer (BD). The gating strategy panel is provided in Supplementary Fig. 11.

**Ca$^{2+}$ measurements of CLL samples.** PBMCs of CLL patients were freshly isolated by ficoll gradient centrifugation were incubated with CD19-APC-Cy7 (HIB19, 1:100) (eBioscience) for 20 min. Cells were washed with PBS w/o Mg$^{2+}$/Ca$^{2+}$ und incubated in 250 μL RPMI 1640 (Invitrogen.) containing 5% FCS (Biochrom), 10 μg/ml FuraRed (Invitrogen) and 0.02% Pluronic F127 (Invitrogen) for 25 min at 30 °C. Cell suspensions were subsequently diluted with 250 μl of medium containing 10% FCS and incubated at 37 °C for another 10 min. Cells were then immediately diluted with 1 ml 0 mM Ca$^{2+}$ Krebs-Ringer solution containing 10 mM Hepes, pH 7.0, 140 mM NaCl, 4 mM KCl, 1 mM MgCl$_2$, 1 mM CaCl$_2$, 10 mM glucose and 0.5 mM EGTA and washed with 1 mL 0 mM Ca$^{2+}$ Krebs-Ringer solution. For the measurements, the cells were diluted with 200 μL 4 mM

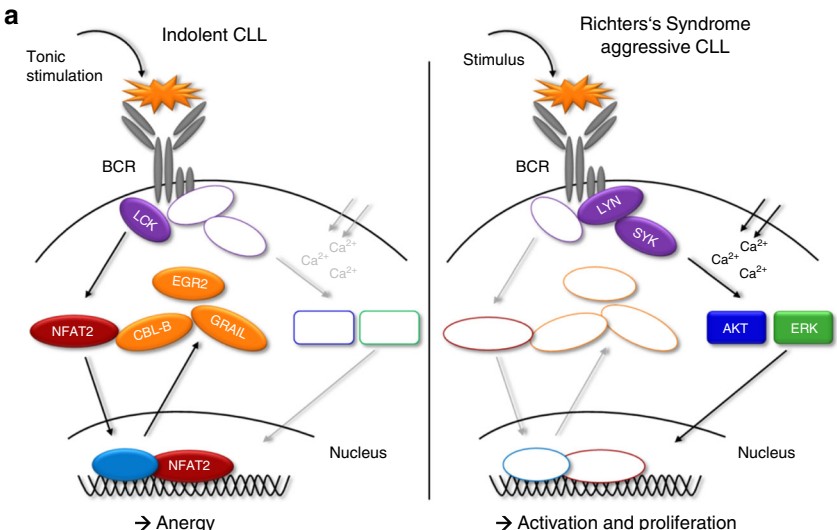

**Fig. 8** Putative mechanism of the role of NFAT2 in the regulation of the anergic phenotype in CLL. (*Left panel*) Tonic stimulation of the BCR in indolent CLL is associated with the recruitment of LCK and the activation of NFAT2, which subsequently translocates to the nucleus to induce the transcription of several anergy-associated genes including *Cbl-b, Grail, Egr2* as well as *Lck*. (*Right panel*) In the absence of NFAT2, BCR stimulation leads to enhanced phosphotyrosine induction and calcium flux culminating in the activation of AKT and ERK and subsequent cell proliferation as observed in patients with aggressive forms of CLL or Richter's syndrome

$Ca^{2+}$ Krebs-Ringer solution and a baseline was recorded for 60 s. $Ca^{2+}$ movement was assessed after stimulation of the cells with 20 μg/ml α-IgM (Southern Biotech). After 4 min of recording, 1 μM Ionomycin was added as a positive control. Increases in free intracellular $Ca^{2+}$ were measured in real-time on a Canto II (BD). To determine the $Ca^{2+}$ flux, the ratio of bound and unbound FuraRed was calculated with FlowJo 8.5.3 software. The gating strategy used for the $Ca^{2+}$ measurements is provided in Supplementary Fig. 11.

**$Ca^{2+}$ measurements of murine B cells**. $1 \times 10^7$ of murine splenic B cells freshly isolated by ficoll gradient centrifugation were incubated with CD19-PECy7 (eBio1D3, 1:100) and CD5-APC (53-7.3, 1:80) (eBioscience) for 20 min. Cells were washed with PBS w/o $Mg^{2+}/Ca^{2+}$ und incubated in 250 μL RPMI 1640 (Invitrogen) containing 5% FCS (Biochrom), 1 μM Indo-1-AM (Invitrogen) and 0.015% Pluronic F127 (Invitrogen) for 25 min at 30 °C. Cell suspensions were subsequently diluted with 250 μL of medium containing 10% FCS and incubated at 37 °C for another 10 min. Cells were then immediately diluted with 1 mL 0 mM $Ca^{2+}$ Krebs-Ringer solution containing 10 mM Hepes, pH 7.0, 140 mM NaCl, 4 mM KCl, 1 mM $MgCl_2$, 1 mM $CaCl_2$, 10 mM glucose and 0.5 mM EGTA and washed with 1 mL 0 mM $Ca^{2+}$ Krebs-Ringer solution. For the measurements, the cells were diluted with 200 μL 0 mM $Ca^{2+}$ Krebs-Ringer solution and a baseline was recorded for 60 s. $Ca^{2+}$ movement was assessed after stimulation of the cells with 10 μg/mL α-IgM (Southern Biotech, Birmingham, AL, USA) followed by the addition of 4 mM $Ca^{2+}$ Krebs-Ringer solution to a total of 2 mM $Ca^{2+}$ after 2 min of recording. After 6 min of recording, 1 μM Ionomycin was added as a positive control. Increases in free intracellular $Ca^{2+}$ were measured in real-time on a LSR II (BD). To determine the $Ca^{2+}$ flux, the ratio of bound and unbound Indo-1-AM was calculated with FlowJo 8.5.3 software. The gating strategy used was identical with the one applied for human CLL $Ca^{2+}$ measurements.

**Real-time RT-PCR**. RNA was isolated from cell pellets using RNeasy plus mini kit (QIAGEN, 74134) followed by reverse transcription (SuperScript II, Invitrogen). Quantitative PCR was performed using SYBR Select Mastermix (Invitrogen) in a LC480 Lightcycler (Hoffmann-La Roche). RT² qPCR™ primer assays (mouse: *Cbl*-b, *Grail*, *Lck*, *Egr2*, *Cdkn2a*, *Trp53*, *β-Actin* & *Gapdh*; human: *LCK*) were purchased from QIAGEN. The mRNA expression ratio was calculated to *β-Actin* or *Gapdh*. Primers for mouse *Prdm1* and human PBMC and lymph node samples were synthesised by Sigma Aldrich (Supplementary Table 2).

**Western blot**. Protein lysates were isolated from cell pellets with RIPA buffer with PI cocktail (both Thermo Scientific) for 30 min on ice followed by 20 min of incubation at 4 °C on a shaker. The supernatant was used for SDS-PAGE followed by western blot analysis. Antibodies for NFAT2 (7A6, BD, 1:2,000), NFAT1 (D43B1, 1:2,000), Actin (8H10D10, 1:5,000) (BD), LYN & P-LYN (Tyr507) (1:1,000), SYK (D1I5Q, 1:1,000) & P-SYK (Tyr519/520) (C87C1, 1:1,000), LCK and P-LCK (Tyr505) (1:1,000), ERK1/2 (3A7, 1:1,000) & P-ERK1/2 (Thr202/Tyr204 Thr185/Tyr187, 1:1,000), GAPDH (D16H11, 1:2,000), Cofilin (D3F9, 1:2,000) (all from Cell signalling) and P-LCK (Tyr394) (GeneTex) were used. IRDye 680RD (1:80,000) and IRDye 800CW (1:20,000) secondary antibodies (LI-COR Bioscience) were used for detection in a LI-COR Odyssey imaging reader. Data was displayed within linear signal ranges. For quantification the Integrated Intensity (K counts mm²) was used and the ratios of target to GAPDH or Actin signals were calculated. Comparative analyses of multiple blots were performed by normalisation to control standards on every individual blot.

**Immunohistochemistry and microscopy**. All organs were fixed in 4% formalin and embedded in paraffin. For histology, 3–5-μm-thick sections were cut and stained with haematoxylin and eosin (H&E). Immunohistochemistry was performed on an automated immunostainer (Ventana Medical Systems, Tucson, AZ, USA) according to the company's protocols for open procedures. The slides were stained with the following antibodies: CD3 (SP7, 1:400) (DCS Innovative Diagnostik-Systeme GmbH u. Co. KG), B220 (RA3-6B2, 1:50) (BD), CD79a (JCB117, 1:50) (DakoCytomation) and Ki-67 (SP6, 1:100) (DCS Innovative Diagnostik-Systeme GmbH u. Co. KG) and LCK (Cell Signalling, 1:100). Appropriate positive and negative controls were used to confirm the adequacy of the staining. Pictures were taken with the Axio Imager A1 (Zeiss) and the ProgRes® C10plus (JenOptik) camera and analysed with ImageAccess 6 Release 07.4 (Imagic Bildverarbeitungs AG).

**Microarray analysis**. Splenocytes of 5 month old mice were isolated by ficoll gradient centrifugation. Splenic $CD19^+CD5^+$ CLL cells were sorted by FACS Aria (BD). RNA was isolated with miRNeasy Kit (Qiagen, 217004) and cDNA was prepared with the Affymetrix WT Sense Labelling Kit (Thermo Fisher Scientific, 900670) according to the manufacturer's instructions. cDNA was hybridised on an Affymetrix MoGene-1.0-ST-v1. The arrays were scanned using an Affymetrix GeneChip Scanner 3000. Data was edited with the AGCC 3.0 software and normalised with GC-RMA (Robust Multichip Average) values. To compare differentially expressed transcripts in wild-type and NFAT2-deficient CLL cells, a linearised model was generated. Expression profiles of different probe sets were determined and *t*-tests with standard errors were calculated by using the empirical Bayes method. *P*-values were corrected by using the Benjamini-Hochberg procedure. Data was obtained in collaboration with the Microarray Facility of the University of Tübingen. The gene expression microarray data reported in this paper have been deposited in the NCBI Gene Expression Omnibus (GEO) database http://www.ncbi.nlm.nih.gov/geo/ with the GEO accession number GSE86210.

**Immunofluorescence and co localisation experiments**. The peripheral blood of *TCL1 Nfat2⁻/⁻* and *TCL1 Nfat2⁺/⁺* mice was lysed with ammonium chloride buffer (0.150 mM $NH_4Cl$, 0.1 mM EDTA, 0.150 mM $KHCO_3$) for 10 min on ice. Cells were either stimulated with 20 μg/ml αIgM F(ab')₂ fragments (Jackson Immuno-Research) for the indicated time points or left untreated. Cytospins were prepared and cells were fixed with acetone for 3 min and frozen overnight at −80 °C. For immunofluorescence, cytospins were treated with 0.2% Triton-X and incubated with primary antibodies overnight. Staining was performed with goat anti-mouse-FITC Ab (Dako, Agilent technologies). For colocalisation experiments the Duolink® kit (Sigma Aldrich, DUO92101) was used according to the manufacturer's instructions. Antibodies for CD79a (Abbexa, 1:100), LYN (clone 11A7, Abcam, 1:100) or CD79a (NSJ, 1:200) and LCK (Cell signalling, 1:50 and a conventional DAPI staining were used for immunofluorescence and co localisation analysis. For negative controls a staining without the primary antibody was performed. Pictures were taken with the Apotome microscope (Zeiss) and analysed with the AxioVision 4.8 software (Zeiss).

**ChIP experiments**. CLL cells were isolated as described above and were stimulated for 16 h with PMA (20 ng/ml, Sigma Aldrich) and ionomycin (0.75 μg/ml, Sigma Aldrich). Cells were fixed with 1% formaldehyde for 5 min and fixation was stopped by the addition of 10% glycine. The chromatin immunoprecipitation was performed using the iDEAL ChIP seq kit (Diagenode, C01010055) according to the manufacturer's instructions. The obtained chromatin was sheared to yield 200–500 bp DNA fragments with the focused ultrasonicator M220 (Covaris) and the magnetic immunoprecipitation was performed using the following antibodies: 10 μg NFAT2 (7A6) (Abcam) and 2 μg IgG (DA1E) (Cell signalling). With the immunoprecipitated DNA, a qPCR was performed using SYBR Select Mastermix (Invitrogen) in a LC480 Lightcycler (Hoffmann-La Roche). The number of immunoprecipitated regions for each target gene (CD40L and LCK) was calculated using different dilutions of input DNA. The primers used for the CD40L and LCK promotor region were synthesised by Sigma Aldrich (Supplementary Table 3).

**Statistical analysis**. For statistical analysis GraphPad Prism 6 was used. Mean values and S.E.M. are shown. The 95% confidence level was used and *P*-values were calculated with an unpaired two-tailed Student´s *t*-test or an unpaired two-tailed Welch's *t*-test in the case of normally distributed data. Significance of not normally distributed data was either calculated with a paired two-tailed Wilcoxon matched-pairs signed-rank test or a two-tailed unpaired Mann–Whitney test. An unpaired analysis of variance (ANOVA) was used to analyse the differences among group means. Significance of survival data was calculated by using a Log-rank (Mantel–Cox) test and a Gehan–Breslow–Wilcoxon test.

**Data availability**. Data supporting the findings of this study are available within the article and its Supplementary Information files and from the corresponding author upon reasonable request. The gene expression microarray data reported in this paper have been deposited in the NCBI Gene Expression Omnibus (GEO) database http://www.ncbi.nlm.nih.gov/geo/ with the GEO accession number GSE86210.

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

## Acknowledgements

This work was supported by the DFG grant MU 3340/1-1 (M.R.M.), the Deutsche Krebshilfe grant 111134 (H.R.S. and M.R.M.), a fortüne grant from the University of Tübingen (M.R.M.) and a scholarship from the Boehringer Ingelheim Fond (J.H.). We thank Carlo M. Croce for providing the Eμ-TCL1 mice and K. Mark Ansel for providing the conditional NFAT2 knockout mice. We thank Alexandra Poljak, Elke Malenke, Manuela Ganser, David Worbs and Bettina Hackl for excellent technical assistance. We thank Irina Bonzheim for collecting the paraffin-embedded lymph node biopsies from patients and RNA isolation.

## Author contributions

M.M., J.S.H., A.R.F., S.W. and M.R.M. designed and performed the experiments, analysed the data and wrote the manuscript. F.M.T., M.G., S.B. J.L. and S.J.S. performed experiments under the supervision of M.M. and M.R.M. U.K. and L.Q.-M. performed and analysed immunohistochemical stains of murine tissue sections. H.-G.R., H.R.S., H.-G.K., M.H., A.K., L.K. and A.R. designed research, analysed the data and provided important advice.

## Additional information

**Competing interests:** The authors declare no competing financial interests.

