## [Peer Review File · Nature Communications]

Reviewers' comments:

Reviewer #1 (Remarks to the Author):

The manuscript by Märklin and collaborators reports a novel animal model of CLL showing that genetic ablation of the transcription factor NFAT2 leads to the loss of the anergic phenotype and subsequent disease acceleration in vivo in the Eu-TCL1 mouse model of CLL.

The rationale of the study is well grounded and the aim of the study commendable. At present, a number of concerns arise on the interpretation of part of the results and on the design of some experiments as described below.

MAJOR COMMENTS

As a general comment, the overexpression of NFAT in CLL has been shown in relation to anergic functional features of CLL and only indirectly associated with the clinical outcome (indolent vs aggressive forms) though it is true that the indolent forms of the leukemia tend to be (but not exclusively) more often anergic. This concept has been overlooked and oversimplified by the authors that seem to ground all their work on the assumption that NFAT overexpression = indolent CLL and they use the 2 terms without distinction, creating confusion in the reader. In addition, this apparent misunderstanding seems to lead to a not entirely correct design of some experiments, e.g. limiting some of the analyses only to indolent CLL (see e.g. beginning of page 5 where a cohort of only indolent cases have been analyzed for NFAT expression, Fig 1a-c). That assumption also creates a great confusion in the wording utilized throughout the text where the term "indolent" is used inappropriately; see e.g. page 5, end of first paragraph where the authors mention "exhibit important features of indolent disease" while they mean "exhibit anergic features" as then shown in figure 1d). In summary, the authors should refer to NFAT overexpression as a feature of anergy in CLL cells (as they appropriately do at the end of page 10 where they indeed study modulation of functional features of anergy) and not as a marker of indolency. Secondly, they should study all CLL cases including aggressive cases (ideally associating functional IG unresponsiveness).

Along similar lines, it is not clear how the authors define indolent cases? It would be interesting to report (maybe in a separate table or in the supplementary material) the clinical and biological features (e.g. IGHV mutational status, CD38, need of therapy, length of follow-up) of all patients studied in the paper to help the readers understand.

- The authors should characterize the effects (including the long term follow up) of NFAT2 ablation in normal B cells in WT (non-TCL1-Tg) mice, analyzing all B cell subpopulations and showing survival curves of the same length. Is the transformation occurring only due to the concomitant presence of TCL1 overexpression?

- Page 5, fig 1, it would also be important to show what happens to NFAT 1 (upregulation?) in the NFAT2 KO mice.

- on page 6, fig 2c, the authors report the increase of white blood cells which I would refrain from showing as it is a very inappropriate way of measuring a lymphocytic leukemia. It could be replaced by the Absolute lymphocyte count (if they need it) otherwise the CD19 CD5 count is more than enough.

- Having said that, as the authors showed the Leucocyte count, from fig 2c, it seems that the leucocyte count increases more rapidly in the NFat2 KO mice as compared to the CD19CD5. Is this indicating that also other leucocytes/lymphocytes rather than CD19CD5 B cells are affected by the loss of NFAT2? Along the same line of reasoning, the authors should show (e.g. in fig 1) if and how other cell populations (in particular CD19posCD5neg B cells) are affected in these mice, as part of a typical characterization of a novel mouse model.

- At the beginning of page 8, again the sentence "transformation of indolent CLL" should be replaced with "transformation of CLL" as the disease in TCL1 is characteristic of aggressive CLL. Similarly on page 11, in the sentence "Indolent CLL cells from TCL1 mice...." the term "indolent" should be

replaced by "anergic".

- At the beginning of page 9, the experimental design creates some confusion. On the one hand, the authors show that a number of anergy related genes are differentially expressed in mice with intact or ablated NFAT2 i.e. downregulated in NFAT2 KO and conversely expressed or upregulated in NFAT-intact mice. Why are the authors then stimulating through the BCR the CLL cells from TCL1 NFAT-intact mice that should be anergic (thus non-responsive) and should already have high levels of those genes? Shouldn't be enough to incubate the "anergic" cells with CsA or VIVIT to show downmodulation of the genes? This experiments should also serve as control for fig 5e-h.

- At the bottom of page 9, the authors should acknowledge that LCK expression in CLL cells has been already repeatedly reported in the literature dating back to 1998.

- On page 11, the sentence "Indolent human CLL is characterized by antigen-independent cell-autonomous signaling with constitutive activation of the BCR signaling cascade45 while aggressive forms of the disease display an enhanced response to external BCR stimuli with consecutive stimulation of downstream targets6." is definitely not reflecting the original manuscript on autonomous signalling neither the subsequent literature. The current knowledge in the field is that all CLL are endowed with autonomous signalling. Actually, any reference to autonomous signalling is not really necessary for this work and it would be better to eliminate this reference tout court as you do not need it as rationale neither for discussion of the results.

- On page 11, the sentence "While anergic CLL cells displayed no evidence of SYK phosphorylation, BCR stimulation clearly induced SYK phosphorylation and activation in the Nfat2-deleted cohort" should be clarified as it is not clear if the absence of SYK phosphorylation in Anergic CLL cells was before or after IG stimulation.

- In the case of Richter transformation with loss of B220, how is the expression of CD5? Is it maintained or lost?

-In fig 6b-c, the authors show Jurkat cells (why not a B cell line?) the lane with IG that should serve as control appear to be quite positive. How can you explain it?

-In fig 6e, it is not reported what type of CLL has been studied? Again only indolent cases? What do we know about their functional response to IG stimulation/Anergy?

-In fig 7, the experiments have been performed by comparing RS with indolent CLL so all differences are difficult to be entirely understood. The authors should compare also aggressive CLL cases vs RS. Along the same line, the authors cannot state (as they do on page 12 - "discussion") that "aggressive forms of CLL or Richter´s syndrome" as they did not study aggressive CLL and RS cannot be considered as aggressive CLL that is a distinct clinical entity (to be more precise one should even consider the so called "Accelerated CLL" in addition to "aggressive CLL" to really be exhaustive on the subject....).

- On page 13, in the sentence "We detected overexpression and constitutional activation in virtually all human CLL samples analyzed." the genes/molecules that are overexpressed are missing. Having said that, the sentence should be modified as the authors did not test an unbiased selection of CLL. Their results apply only to indolent CLL.

MINOR COMMENTS

- Line 4 of the introduction, it would be correct to add also ofatumumab among the antibodies utilized in CLL

- same line: the term BCR inhibitor is not appropriate and not used anymore. Better to simply use "kinase inhibitors"

- Line 10 of the introduction and throughout the text: "IgVH" should be replaced by the international acronym "IGHV"

- Line 11 and throughout the text: ZAP-70 should be replaced by the international gene acronym ZAP70

- page 4, four lines from the bottom, "overexpression in indolent CLL" should be replaced by

"Overexpression in anergic CLL" as discussed above.

- bottom of page 5: I would refrain from immediately stating "a significantly more aggressive disease course" as in this paragraph the authors describe only the increase in cell count. The difference in survival comes in the subsequent paragraph where indeed they can then make the point of a more aggressive disease.

Reviewer #2 (Remarks to the Author):

The manuscript by Märklin and al. entitled "NFAT2 is a critical regulator of the anergic phenotype in chronic lymphocytic leukemia" analyses the implication of NFAT2 and its targets in the indolent profile of CLL using both a murine model and patient samples.

Experimental design includes the generation of a B-cell specific invalidation of NFAT2 in the context of the E μ -TCL1 murine CLL model. Results report also transcriptomal analysis of NFAT2 target genes in the context of the anergic status of the cells as compared to more aggressive transformation of the disease.

The reported results are based on an elegant approach. They are combining analysis of the novel Nfat2^{-/-} mice developed in the context of the E μ -TCL1 transgenic model, an adequate murine CLL counterpart, and, an investigation of the regulatory role of NFAT in the anergic phenotype of the disease. Based on these designs the authors raise three major conclusions:

- 1) B-cell specific invalidation of Nfat2 in the E μ -TCL1 mice leads to an accelerated and more aggressive development of the disease resembling those evidenced during Richter transformation.
- 2) Ablation of Nfat2 results in the specific downregulation of several important NFAT2 target genes involved in lymphocytic anergy such as Cbl-b, Grail, Egr2 and Lck and in cell cycle progression.
- 3) A differential phosphorylation status of two Src kinases, Lyn and Lck is evidenced both in the murine model and in CLL or Richter samples that seems to be correlated with the transformation.

Although the analysis based on the novel murine model is interesting, some of the conclusions should be reinforced and some concerns remain to be clarified.

Major points:

- 1) The experiments with human samples are always comparing normal B cells, CLL cells and eventually Richter transformed cells (Fig 1, 6 and 7). Taking into account the importance of NFAT2 expression for the development of CD5⁺ CD19⁺ "B1a-like" cells and its differential expression between lymphoid subpopulations, it is of importance to compare experimentally equivalent subpopulations between healthy controls, CLL samples and Richter transformations and to provide information on the purification procedures as well as phenotypic features of the populations examined before any conclusion.
- 2) Flow cytometry analysis of the three murine models (Fig 2 and 3) shows as already documented an increase of the CD19⁺ CD5⁺ subpopulation in E μ -TCL1 transgenic mice and further increase upon ablation of Nfat2. However, while, as expected, E μ -TCL1 transgenics show modification of this population only, the B-cell specific ablation of Nfat2 results in a large decrease of other cell populations that should not be directly impacted such as the CD19⁻ CD5⁺ population. An exhaustive analysis of the various lymphoid subpopulations in which NFAT2 has a relevant impact should be provided. This point is of high importance since the evaluation of the functional impact of the ablation is afterward performed as a result on the whole population and not in a particular subtype. Of note Figures 2c and d indicate clearly that Nfat2 ablation impacts not only CD5⁺CD19⁺ cells but also other white blood cells and similar modifications are observed in spleens in Figure 3d.
- 3) Relationship between accelerated tumor cell turnover (notably annexinV labeling Fig 2 e and f), cell

cycle progression and aggressiveness of the disease should be more documented (comparison with NSG transplanted animals? Upon transformation to aggressive disease?)

4) The authors conclude on the absence of evidence for a relevant gene dosage effect. However in the E μ -TCL1 context, Kaplan-Meier plots in Figure 3a and supplementary Figure 1 (TCL1 Nfat $+/+$) indicate different 50% survival (325 versus 297 or 291 days). If the difference between $+/+$ and $+/-$ animals is 325 versus 267 and not 291 versus 267 would this increase generate significance? Values should be verified and compiled in a single Figure.

5) Conclusions raised from the experiments in Figure 6g are unclear and should be more controlled. Characterization of a colocalization of the Src kinases Lyn and Lck with the antigen receptor is based on a reporter and amplification system. The minimal amount of proteins necessary to visualize the colocalization is not assessed and the experiment not quantitative. Since TCL1Nfat2 $-/-$ cells express very low levels of Lck as compared to TCL1 Nfat2 $+/+$ cells the authors should provide cell staining indicating the residual levels of expression of Lck in the cells before concluding on the absence of the colocalization of the protein with CD79a. Also, Lyn kinase recruitment to the complex seems different between the two mouse models and the antibodies used are not specific of the activated or closed forms of the proteins. Therefore it is difficult to conclude on the absence of recruitment under resting conditions as compared to BCR engagement. Staining for protein expression and activation should be provided in order to confirm the conclusions.

6) The implication of NFAT2 in B CLL cells anergy is documented through the analysis of the phosphorylation state of Lyn and Lck (Fig 7). The authors argue for a balance between the inhibitory phosphorylation of Lyn and an activated phosphorylation status of Lck in anergic cells. Both phosphorylation status are lowered in TCL1 Nfat2 $-/-$ cells. However since p394-Lck is very low in TCL1 Nfat2 $-/-$ cells one should expect that the activation is then due to the p397 Lyn active form. Due to the interplay of the two kinases and the shortage of documentation on the role of Lck in B cells the authors should clarify this important point. A quantitative analysis of the relative level of expression of Lyn and Lck should be performed altogether with the analysis of both inactive and active phosphorylation forms of the kinases in the two mice model upon BCR stimulation. p397 Lyn and p505 Lck status are clearly neglected. Similarly activation status of Lck would have been of interest in human samples in Fig 7g.

Minor Points :

- 1) It would be of interest to visualize the relative level of NFAT2 protein expression in NFAT2 $+/+$ mice in order to correlate their calcium mobilization with the other mice models expressing high NFAT2 (TCL1 NFAT2 $+/+$) or not (TCL1 NFAT2 $-/-$) (Fig 1g)
- 2) Page 6: numbers of days should be identical to those in figures
- 3) Page 7: Fig 4, Supplementary Figure 2 should be corrected.
- 4) Page 9: the sentence Irrelevant IgG and an anti acetyl-histone H3 antibody served as positive and negative controls, respectively should be modified

Reviewer #3 (Remarks to the Author):

In this manuscript Marklin et al investigate the role and significance of the transcription factor NFAT2 in chronic lymphocytic leukemia (CLL) starting from the well known notion that the constitutive activation of Nfat2 is a hallmark of the anergic phenotype in CLL. They essentially use the Tcl1 CLL mouse model but also present data (albeit less numerous and impressive) on human samples. Their main result is that the ablation of Nfat2 in Tcl1 mice leads to the loss of the anergic CLL phenotype and favours the transformation of an indolent disease into an aggressive one which is reminiscent of the Richter transformation in human CLL. In other words the Authors molecularly

characterize the role of a transcription factor which was already known to pinpoint the anergic phenotype of CLL, but whose molecular definition was not known, certainly not to the detailed degree shown by the Authors. Furthermore the Authors find a gene expression signature of anergic CLL cells which consists of several NFAT2-dependant genes and may have a number of implications. The take home message is that NFAT2 is a crucial regulator of the anergic phenotype in CLL whose loss causes the transition from an indolent to an aggressive disease.

These findings are of interest but raise a number of questions. The first is: what regulates the regulator in CLL ? To be more specific why NFAT2 at a certain point of the natural history of CLL is or becomes inactivated and what leads to its inactivation in human beings, considering that this inactivation may lead to the aggressive transformation ? A second major question is related to the fact that a number of genes have already been found or suggested to be involved in the transformation of CLL into Richter's syndrome. How do these genes relate to NFAT2? Is NFAT2 sort of a hub ? Are the already described genes capable of influencing or downregulating NFAT2 so that its role of "anergic brake" is lost ? Though admittedly it is not easy to have at hand many cases of Richter's syndrome, the question is whether the number of human samples is sufficient to reproduce in humans the mouse data. Also what happens to NFAT2 in the cells of patients who are treatment-unresponsive, aggressive but are NOT in histologic transformation? These patients are numerous.

These questions should be at least asked otherwise the discussion remains a mere repetition of the results.

On a more literary ground I find the Introduction and especially the Discussion too long and verbose.

REVIEWERS' COMMENTS:

Reviewer #1 (Remarks to the Author):

The authors have greatly improved the manuscript by performing additional experiments as suggested helping to clarify some issues and eliminate concerns.

Reviewer #2 (Remarks to the Author):

The reviewer acknowledges the revised version of the manuscript by Märlin et al . The authors designed important and informative new experiments in order to answer the numerous comments addressed by the reviewers. The experiments confirmed important findings on the TCL1Nfat2^{-/-} mice notably that NFAT2 ablation leads to an earlier and more aggressive development of the disease resembling those evidenced during Richter transformation. It also emphasizes the role of NFAT2 in the regulation of anergy-linked genes in line with a similar regulation observed in CLL patients at different stages. However, several responses are still elusive and do not address the comments previously raised.

1) The confusion raised in the initial manuscript between the anergic phenotype of several CLL cells and the indolent clinical outcome with only a focus on NFAT and Lck expression has been addressed during edition of the manuscript but is still only very limited on an experimental point of view. It is important that the authors provide direct evidences of the activation status of the cells for patients with "indolent" or "aggressive" outcome (Calcium or phosphorylation of effectors and not only of Lck). Moreover, the papers indicated in reference to this notion do not conclude precisely on this specific link between anergy and indolent disease (Page 4 introduction, p13 Discussion ref 9, 11, 23, 24, 29, 30). This point should be appropriately addressed and clarified in CLL patients with different profiles used in the study.

2) There is still some controversial explanation between the flow cytometry data in figure 2 and 3 and explanation in the text on page 6 (increase of the CD19⁺ CD5⁻ during later disease stages with internalization of CD5 is not seen in figure 2). Additionally, explanation on pronounced tumor infiltrate is not shown and sufficient to explain the dot plots in figures 2 and 3 for CD19⁻ CD5⁺ or in figure 4 while sup figure 3 shows no alteration by NFAT ablation for T cell population (CD3⁺). Regarding the important role of NFAT2 in various lymphoid lineages these findings should be more precisely addressed and clarified in the manuscript (pages 6 and 8).

3) Explanation on the recruitment of Lyn versus Lck to the CD79a is quite misleading since in TCL1 Nfat2^{-/-} resting cells both Lck and Lyn are down regulated (sup Figure 7) and Lyn is not recruited (resting cells) anymore or weakly detected (upon stimulation) (Figure 6). Since technical problems do not allow detection of the activated form of Lyn but Y507 P-Lyn is downregulated in TCL1 Nfat2^{-/-} mice as compared to TCL1 Nfat2^{+/+} (Figure 7d) it would be important to provide additional comments using either P394 Lck or P Lyn507 in Figure 6h or at least use the similar antibodies for patient samples in figure 7g before concluding on the replacement of one protein by the other.

The reviewer considers that all the other requests have been addressed.

Reviewer #3 (Remarks to the Author):

The Authors have satisfactorily answered the queries I had raised in my original Review.

Reviewer #1:

Remarks to the author:

The manuscript by Märklin and collaborators reports a novel animal model of CLL showing that genetic ablation of the transcription factor NFAT2 leads to the loss of the anergic phenotype and subsequent disease acceleration in vivo in the Eu-TCL1 mouse model of CLL. The rationale of the study is well grounded and the aim of the study commendable. At present, a number of concerns arise on the interpretation of part of the results and on the design of some experiments as described below.

Major Comments:

1. As a general comment, the overexpression of NFAT in CLL has been shown in relation to anergic functional features of CLL and only indirectly associated with the clinical outcome (indolent vs aggressive forms) though it is true that the indolent forms of the leukemia tend to be (but not exclusively) more often anergic. This concept has been overlooked and oversimplified by the authors that seem to ground all their work on the assumption that NFAT overexpression = indolent CLL and they use the 2 terms without distinction, creating confusion in the reader. In addition, this apparent misunderstanding seems to lead to a not entirely correct design of some experiments, e.g. limiting some of the analyses only to indolent CLL (see e.g. beginning of page 5 where a cohort of only indolent cases have been analyzed for NFAT expression, Fig 1a-c). That assumption also creates a great confusion in the wording utilized throughout the text where the term "indolent" is used inappropriately; see e.g. page 5, end of first paragraph where the authors mention "exhibit important features of indolent disease" while they mean "exhibit anergic features" as then shown in figure 1d). In summary, the authors should refer to NFAT overexpression as a feature of anergy in CLL cells (as they appropriately do at the end of page 10 where they indeed study modulation of functional features of anergy) and not as a marker of indolency. Secondly, they should study all CLL cases including aggressive cases (ideally associating functional IG unresponsiveness). Along similar lines, it is not clear how the authors define indolent cases? It would be interesting to report (maybe in a separate table or in the supplementary material) the clinical and biological features (e.g.

Universitätsklinikum Tübingen

Anstalt des öffentlichen Rechts
Sitz Tübingen
Geissweg 3 - 72076 Tübingen
Telefon (07071) 29-0
www.medizin.uni-tuebingen.de
Steuer-Nr. 86156/09402
USt.-ID: DE 146 889 674

Aufsichtsrat

Hartmut Schrade (Vorsitzender)

Vorstand

Prof. Dr. Michael Bamberg (Vorsitzender)
Gabriele Sonntag (Stellv. Vorsitzende)*
Prof. Dr. Karl Ulrich Bartz-Schmidt
Prof. Dr. Ingo B. Autenrieth
Klaus Tischler

Banken

Baden-Württembergische Bank Stuttgart
(BLZ 600 501 01) Konto-Nr. 7477 5037 93
IBAN: DE41 6005 0101 7477 5037 93
SWIFT-Nr.: SOLADEST
Kreissparkasse Tübingen
(BLZ 641 500 20) Konto-Nr. 14 144
IBAN: DE79 6415 0020 0000 0141 44

IGHV mutational status, CD38, need of therapy, length of follow-up) of all patients studied in the paper to help the readers understand.

Response: We agree with the reviewer that our assumption “NFAT2 overexpression =indolent CLL” was not justified by the data provided in the original version of the manuscript. We have now included data on the expression levels of NFAT2 in aggressive cases of CLL and demonstrate that NFAT2 overexpression on the mRNA- and protein level is significantly stronger in cases with indolent disease. The discrimination between indolent and aggressive CLL was based on commonly used clinical and prognostic parameters (Binet stage, requirement for treatment, median time to first treatment, number of treatments received, IGHV mutational status, presence of high risk genetic aberrations). The newly included Supplementary Table 1 includes all parameters of the CLL patients used in this study and provides details on how the discrimination in indolent and aggressive cases was made. We also agree with the reviewer that our statement "exhibit important features of indolent disease" (page 5, end of the first paragraph) is misleading and changed it to " exhibit important features of anergic disease which is typically associated with an indolent clinical course" as suggested.

2. The authors should characterize the effects (including the long term follow up) of NFAT2 ablation in normal B cells in WT (non-TCL1-Tg) mice, analyzing all B cell subpopulations and showing survival curves of the same length. Is the transformation occurring only due to the concomitant presence of TCL1 overexpression?

Response: The Kaplan Meier Plot in Fig. 3a now includes survival data on *Nfat2*^{+/+} mice with intact NFAT expression and *Nfat2*^{-/-} mice with B cell-specific deletion of NFAT2 without expression of the TCL1 transgene. Both cohorts exhibited no evidence of disease during the 400 day follow-up period. All animals were alive at the end of the study. We have now also analyzed in detail other lymphoid subpopulations of the different mouse strains in the peripheral blood and in the spleen by flow cytometry (follicular B cells, marginal zone B cells, memory B cells, T cells) (Supplementary Fig. 3).

Universitätsklinikum Tübingen

Anstalt des öffentlichen Rechts
Sitz Tübingen
Geissweg 3 - 72076 Tübingen
Telefon (07071) 29-0
www.medizin.uni-tuebingen.de
Steuer-Nr. 86156/09402
USt.-ID: DE 146 889 674

Aufsichtsrat

Hartmut Schrade (Vorsitzender)

Vorstand

Prof. Dr. Michael Bamberg (Vorsitzender)
Gabriele Sonntag (Stellv. Vorsitzende)*
Prof. Dr. Karl Ulrich Bartz-Schmidt
Prof. Dr. Ingo B. Autenrieth
Klaus Tischler

Banken

Baden-Württembergische Bank Stuttgart
(BLZ 600 501 01) Konto-Nr. 7477 5037 93
IBAN: DE41 6005 0101 7477 5037 93
SWIFT-Nr.: SOLADEST
Kreissparkasse Tübingen
(BLZ 641 500 20) Konto-Nr. 14 144
IBAN: DE79 6415 0020 0000 0141 44

3. Page 5, fig 1, it would also be important to show what happens to NFAT 1 (upregulation?) in the NFAT2 KO mice.

Response: As suggested by the reviewer, we have now performed an analysis of NFAT1 expression in the different mouse strains used in this study by western blotting which is included as the new Supplementary Figure 1. While NFAT2 expression was completely absent in B cells from NFAT2 KO mice, NFAT1 protein expression in NFAT2 KO mice was comparable with *Nfat2*^{+/+} mice and TCL1 transgenic mice with intact NFAT2 expression.

4. On page 6, fig 2c, the authors report the increase of white blood cells which I would refrain from showing as it is a very inappropriate way of measuring a lymphocytic leukemia. It could be replaced by the Absolute lymphocyte count (if they need it) otherwise the CD19 CD5 count is more than enough.

Response: Figure 2c was modified and includes now only the absolute lymphocyte count while the white blood count was eliminated. Fig. 2d continues to show the accumulation of CD19+ CD5+ B cells.

5. Having said that, as the authors showed the Leucocyte count, from fig 2c, it seems that the leucocyte count increases more rapidly in the NFAT2 KO mice as compared to the CD19CD5. Is this indicating that also other leucocytes/lymphocytes rather than CD19CD5 B cells are affected by the loss of NFAT2? Along the same line of reasoning, the authors should show (e.g. in fig 1) if and how other cell populations (in particular CD19posCD5neg B cells) are affected in these mice, as part of a typical characterization of a novel mouse model.

Response: We agree with the reviewer that the depiction in the original manuscript suggests that other leukocyte populations in addition to the CD19+ CD5+ B cells are also affected by CD19-Cre mediated *Nfat2* deletion. We have now also carefully assessed the accumulation of CD19+ CD5- B cells in the different experimental cohorts.

Universitätsklinikum Tübingen

Anstalt des öffentlichen Rechts
Sitz Tübingen
Geissweg 3 - 72076 Tübingen
Telefon (07071) 29-0
www.medizin.uni-tuebingen.de
Steuer-Nr. 86156/09402
USt.-ID: DE 146 889 674

Aufsichtsrat

Hartmut Schrade (Vorsitzender)

Vorstand

Prof. Dr. Michael Bamberg (Vorsitzender)
Gabriele Sonntag (Stellv. Vorsitzende)*
Prof. Dr. Karl Ulrich Bartz-Schmidt
Prof. Dr. Ingo B. Autenrieth
Klaus Tischler

Banken

Baden-Württembergische Bank Stuttgart
(BLZ 600 501 01) Konto-Nr. 7477 5037 93
IBAN: DE41 6005 0101 7477 5037 93
SWIFT-Nr.: SOLADEST
Kreissparkasse Tübingen
(BLZ 641 500 20) Konto-Nr. 14 144
IBAN: DE79 6415 0020 0000 0141 44

And indeed there appears to be also an increase of the CD19⁺ CD5⁻ population in TCL1 *Nfat2*^{-/-} mice (Supplementary Figure 2a). This effect is most likely due to a downregulation of CD5 in the TCL1 *Nfat2*^{-/-} cohort upon leukemia transformation as shown in Supplementary Figure 2b of the revised version of the manuscript.

6. At the beginning of page 8, again the sentence "transformation of indolent CLL" should be replaced with "transformation of CLL" as the disease in TCL1 is characteristic of aggressive CLL. Similarly on page 11, in the sentence "Indolent CLL cells from TCL1 mice...." the term "indolent" should be replaced by "anergic".

Response: The suggested changes have been made in the new version of the manuscript.

7. At the beginning of page 9, the experimental design creates some confusion. On the one hand, the authors show that a number of anergy related genes are differentially expressed in mice with intact or ablated NFAT2 i.e. downregulated in NFAT2 KO and conversely expressed or upregulated in NFAT-intact mice. Why are the authors then stimulating through the BCR the CLL cells from TCL1 NFAT-intact mice that should be anergic (thus non-responsive) and should already have high levels of those genes? Shouldn't be enough to incubate the "anergic" cells with CsA or VIVIT to show downmodulation of the genes? This experiments should also serve as control for fig 5e-h.

Response: NFAT2 comes out of the nucleus within 15 min after removal from the stimulus, so during the time required to isolate the cells and place them in culture the anergic phenotype can become degraded, such that the baseline level of expression of the genes can be quite variable from one experiment to another and the effects of CsA treatment can be variable and hard to see when the level of expression is low. This is less of a problem when using cells directly ex vivo, which is why we see substantial differences between *Nfat2*^{+/+} and *Nfat2*^{-/-} cells in Figs 5a-d.

Universitätsklinikum Tübingen

Anstalt des öffentlichen Rechts
Sitz Tübingen
Geissweg 3 - 72076 Tübingen
Telefon (07071) 29-0
www.medizin.uni-tuebingen.de
Steuer-Nr. 86156/09402
USt.-ID: DE 146 889 674

Aufsichtsrat

Hartmut Schrade (Vorsitzender)

Vorstand

Prof. Dr. Michael Bamberg (Vorsitzender)
Gabriele Sonntag (Stellv. Vorsitzende)*
Prof. Dr. Karl Ulrich Bartz-Schmidt
Prof. Dr. Ingo B. Autenrieth
Klaus Tischler

Banken

Baden-Württembergische Bank Stuttgart
(BLZ 600 501 01) Konto-Nr. 7477 5037 93
IBAN: DE41 6005 0101 7477 5037 93
SWIFT-Nr.: SOLADEST
Kreissparkasse Tübingen
(BLZ 641 500 20) Konto-Nr. 14 144
IBAN: DE79 6415 0020 0000 0141 44

To overcome this problem, we decided to take advantage of the fact that the anergic phenotype degrades as the cells are being manipulated ex vivo. Our only point in Figures 5e-h was to show that the four genes examined — Cbl-b, Grail, Egr2 and Lck — are in fact transcriptional targets of the calcium/ calcineurin pathway. We therefore stimulated the cells through the BCR to achieve a reproducible level of baseline expression and compared expression in the presence and absence of CsA or FK506 (Figure 5e-h).

8. At the bottom of page 9, the authors should acknowledge that LCK expression in CLL cells has been already repeatedly reported in the literature dating back to 1998.

Response: The fact that Lck expression in CLL cells has been repeatedly reported in the literature is now acknowledged in the manuscript and the respective references are added to the bibliography.

9. On page 11, the sentence "Indolent human CLL is characterized by antigen-independent cell-autonomous signaling with constitutive activation of the BCR signaling cascade⁴⁵ while aggressive forms of the disease display an enhanced response to external BCR stimuli with consecutive stimulation of downstream targets⁶." is definitely not reflecting the original manuscript on autonomous signalling neither the subsequent literature. The current knowledge in the field is that all CLL are endowed with autonomous signalling. Actually, any reference to autonomous signalling is not really necessary for this work and it would be better to eliminate this reference tout court as you do not need it as rationale neither for discussion of the results.

Response: We agree with the reviewer that the reference to autonomous signaling is not required for the discussion of the results of this study and eliminated it from the updated version of the manuscript.

Universitätsklinikum Tübingen

Anstalt des öffentlichen Rechts
Sitz Tübingen
Geissweg 3 - 72076 Tübingen
Telefon (07071) 29-0
www.medizin.uni-tuebingen.de
Steuer-Nr. 86156/09402
USt.-ID: DE 146 889 674

Aufsichtsrat

Hartmut Schrade (Vorsitzender)

Vorstand

Prof. Dr. Michael Bamberg (Vorsitzender)
Gabriele Sonntag (Stellv. Vorsitzende)*
Prof. Dr. Karl Ulrich Bartz-Schmidt
Prof. Dr. Ingo B. Autenrieth
Klaus Tischler

Banken

Baden-Württembergische Bank Stuttgart
(BLZ 600 501 01) Konto-Nr. 7477 5037 93
IBAN: DE41 6005 0101 7477 5037 93
SWIFT-Nr.: SOLADEST
Kreissparkasse Tübingen
(BLZ 641 500 20) Konto-Nr. 14 144
IBAN: DE79 6415 0020 0000 0141 44

10. On page 11, the sentence "While anergic CLL cells displayed no evidence of SYK phosphorylation, BCR stimulation clearly induced SYK phosphorylation and activation in the *Nfat2*-deleted cohort" should be clarified as it is not clear if the absence of SYK phosphorylation in Anergic CLL cells was before or after IG stimulation.

Response: In anergic CLL cells (TCL1 *Nfat2*^{+/+}) an absence of SYK phosphorylation could be detected before and after BCR stimulation. In NFAT2-deleted cells (TCL1 *Nfat2*^{-/-}), an absence of SYK phosphorylation could be detected only before BCR stimulation while IgM treatment induced SYK phosphorylation. The necessary clarifications have been made in the revised version of the manuscript.

11. In the case of Richter transformation with loss of B220, how is the expression of CD5? Is it maintained or lost?

Response: In the TCL1 *Nfat2*^{-/-} cohort we were able to document loss of B220 as well as a downregulation of CD5 upon disease transformation. Supplementary Figure 2b of the revised manuscript contains an analysis of the mean fluorescence intensity (MFI) for CD5 in the different animal cohorts at 12 and 28 weeks of age. While CD5 expression was stable in *Nfat2*^{+/+} and TCL1 *Nfat2*^{+/+} mice, it was substantially downregulated in the TCL1 *Nfat2*^{-/-} cohort at 28 weeks of age.

12. In fig 6b-c, the authors show Jurkat cells (why not a B cell line?) the lane with IG that should serve as control appear to be quite positive. How can you explain it?

Response: The original purpose of the experiment depicted in Figure 6b was to demonstrate that NFAT2 does bind to the Lck promoter and to define thus Lck as potential NFAT2 target. We agree with the reviewer that observations made in Jurkat cells can not necessarily be extended to primary human B cells. We have therefore performed the chromatin immunoprecipitation with primary human CLL cells and could show that NFAT2 does also bind the Lck promoter in this context. This experi-

Universitätsklinikum Tübingen

Anstalt des öffentlichen Rechts
Sitz Tübingen
Geissweg 3 - 72076 Tübingen
Telefon (07071) 29-0
www.medizin.uni-tuebingen.de
Steuer-Nr. 86156/09402
USt.-ID: DE 146 889 674

Aufsichtsrat

Hartmut Schrade (Vorsitzender)

Vorstand

Prof. Dr. Michael Bamberg (Vorsitzender)
Gabriele Sonntag (Stellv. Vorsitzende)*
Prof. Dr. Karl Ulrich Bartz-Schmidt
Prof. Dr. Ingo B. Autenrieth
Klaus Tischler

Banken

Baden-Württembergische Bank Stuttgart
(BLZ 600 501 01) Konto-Nr. 7477 5037 93
IBAN: DE41 6005 0101 7477 5037 93
SWIFT-Nr.: SOLADEST
Kreissparkasse Tübingen
(BLZ 641 500 20) Konto-Nr. 14 144
IBAN: DE79 6415 0020 0000 0141 44

ment replaces the original ChIP experiment with Jurkat cells in the updated version of the manuscript (Figure 6b)

13. In fig 6e, it is not reported what type of CLL has been studied? Again only indolent cases? What do we know about their functional response to IG stimulation/Anergy?

Response: In Figure 6 c-f of the revised version of the manuscript we now provide a differentiation in indolent and aggressive CLL cases. As for NFAT2, the expression of LCK was significantly higher in CLL cells from patients with indolent disease than in CLL cells from patients with aggressive disease. On the other hand, it has been shown by us and others that NFAT2 overexpression correlates with anergic features of CLL cells and their unresponsiveness to BCR/IgM stimulation.

14. In fig 7, the experiments have been performed by comparing RS with indolent CLL so all differences are difficult to be entirely understood. The authors should compare also aggressive CLL cases vs RS. Along the same line, the authors cannot state (as they do on page 12 - "discussion") that "aggressive forms of CLL or Richter's syndrome" as they did not study aggressive CLL and RS cannot be considered as aggressive CLL that is a distinct clinical entity (to be more precise one should even consider the so called "Accelerated CLL" in addition to "aggressive CLL" to really be exhaustive on the subject....).

Response: We now have analyzed patients with indolent and aggressive disease for the expression levels of LCK and phospho-LCK in their CLL cells. We could detect a significant increased expression of LCK and phospho-LCK in CLL cells from patients with indolent disease, while cells from patients with aggressive CLL or Richter's syndrome showed LCK expression comparable to physiological B cells (Fig. 6d-f, Fig. 7 e-g). Here, one has to keep in mind that diagnosis of CLL is commonly made in the peripheral blood without the necessity of a lymph node or bone marrow biopsy while a Richter's transformation is usually confirmed by lymph node or other tissue biopsy which complicates the direct comparison of CLL (indolent/aggressive) and Richter's syndrome with respect to expression of NFAT2 and LCK in most available cases. In Figure 7, we provide data on the expression of NFAT2 and LCK in the lymph nodes

Universitätsklinikum Tübingen

Anstalt des öffentlichen Rechts
Sitz Tübingen
Geissweg 3 - 72076 Tübingen
Telefon (07071) 29-0
www.medizin.uni-tuebingen.de
Steuer-Nr. 86156/09402
USt.-ID: DE 146 889 674

Aufsichtsrat

Hartmut Schrade (Vorsitzender)

Vorstand

Prof. Dr. Michael Bamberg (Vorsitzender)
Gabriele Sonntag (Stellv. Vorsitzende)*
Prof. Dr. Karl Ulrich Bartz-Schmidt
Prof. Dr. Ingo B. Autenrieth
Klaus Tischler

Banken

Baden-Württembergische Bank Stuttgart
(BLZ 600 501 01) Konto-Nr. 7477 5037 93
IBAN: DE41 6005 0101 7477 5037 93
SWIFT-Nr.: SOLADEST
Kreissparkasse Tübingen
(BLZ 641 500 20) Konto-Nr. 14 144
IBAN: DE79 6415 0020 0000 0141 44

of patients with indolent CLL and patients with confirmed Richter's transformation. We now have also performed an immunohistochemical stain for LCK of the lymph node sections of both patient groups and confirm the almost complete downregulation of LCK in patients with Richter's syndrome (Figure 7h).

15. On page 13, in the sentence "We detected overexpression and constitutional activation in virtually all human CLL samples analyzed." the genes/molecules that are overexpressed are missing. Having said that, the sentence should be modified as the authors did not test an unbiased selection of CLL. Their results apply only to indolent CLL.

Response: Taking into account our differential analysis of cases with indolent and aggressive CLL, we have modified the text to "We detected overexpression of NFAT2 and overexpression as well as constitutional activation of LCK in virtually all indolent CLL samples analyzed" in the revised version of the manuscript.

Minor Comments:

1. Line 4 of the introduction, it would be correct to add also ofatumumab among the antibodies utilized in CLL.

Response: Ofatumumab is now added to the list of antibodies in the revised version of the manuscript.

2. Same line: the term BCR inhibitor is not appropriate and not used anymore. Better to simply use "kinase inhibitors".

Response: The term BCR inhibitor was replaced by the term kinase inhibitor.

3. Line 10 of the introduction and throughout the text: "IgVH" should be replaced by the international acronym "IGHV".

Universitätsklinikum Tübingen

Anstalt des öffentlichen Rechts
Sitz Tübingen
Geissweg 3 - 72076 Tübingen
Telefon (07071) 29-0
www.medizin.uni-tuebingen.de
Steuer-Nr. 86156/09402
USt.-ID: DE 146 889 674

Aufsichtsrat

Hartmut Schrade (Vorsitzender)

Vorstand

Prof. Dr. Michael Bamberg (Vorsitzender)
Gabriele Sonntag (Stellv. Vorsitzende)*
Prof. Dr. Karl Ulrich Bartz-Schmidt
Prof. Dr. Ingo B. Autenrieth
Klaus Tischler

Banken

Baden-Württembergische Bank Stuttgart
(BLZ 600 501 01) Konto-Nr. 7477 5037 93
IBAN: DE41 6005 0101 7477 5037 93
SWIFT-Nr.: SOLADEST
Kreissparkasse Tübingen
(BLZ 641 500 20) Konto-Nr. 14 144
IBAN: DE79 6415 0020 0000 0141 44

Response: The international acronym IGVH is now used throughout the manuscript as suggested.

4. Line 11 and throughout the text: ZAP-70 should be replaced by the international gene acronym ZAP70.

Response: The international acronym ZAP70 is now used throughout the manuscript as suggested.

5. Page 4, four lines from the bottom, "overexpression in indolent CLL" should be replaced by "Overexpression in anergic CLL" as discussed above.

Response: The respective change has been made in the new version of the manuscript.

6. Bottom of page 5: I would refrain from immediately stating "a significantly more aggressive disease course" as in this paragraph the authors describe only the increase in cell count. The difference in survival comes in the subsequent paragraph where indeed they can then make the point of a more aggressive disease.

Response: The statement "a significantly more aggressive disease course" was replaced with "a significant acceleration of leukemia development".

Reviewer #2:

Remarks to the author:

The manuscript by Märklin and al. entitled "NFAT2 is a critical regulator of the anergic phenotype in chronic lymphocytic leukemia" analyses the implication of NFAT2 and its targets in the indolent profile of CLL using both a murine model and patient samples. Experimental design includes the generation of a B-cell specific invalidation of NFAT2 in the context of the E μ -TCL1 murine CLL model. Results report also transcriptomal analysis of NFAT2 target genes in the context of the anergic status of the cells as compared to more aggressive trans-

Universitätsklinikum Tübingen

Anstalt des öffentlichen Rechts
Sitz Tübingen
Geissweg 3 - 72076 Tübingen
Telefon (07071) 29-0
www.medizin.uni-tuebingen.de
Steuer-Nr. 86156/09402
USt.-ID: DE 146 889 674

Aufsichtsrat

Hartmut Schrade (Vorsitzender)

Vorstand

Prof. Dr. Michael Bamberg (Vorsitzender)
Gabriele Sonntag (Stellv. Vorsitzende)*
Prof. Dr. Karl Ulrich Bartz-Schmidt
Prof. Dr. Ingo B. Autenrieth
Klaus Tischler

Banken

Baden-Württembergische Bank Stuttgart
(BLZ 600 501 01) Konto-Nr. 7477 5037 93
IBAN: DE41 6005 0101 7477 5037 93
SWIFT-Nr.: SOLADEST
Kreissparkasse Tübingen
(BLZ 641 500 20) Konto-Nr. 14 144
IBAN: DE79 6415 0020 0000 0141 44

formation of the disease. The reported results are based on an elegant approach. They are combining analysis of the novel *Nfat2*^{-/-} mice developed in the context of the E μ -TCL1 transgenic model, an adequate murine CLL counterpart, and, an investigation of the regulatory role of NFAT in the anergic phenotype of the disease. Based on these designs the authors raise three major conclusions:

- 1) B-cell specific invalidation of *Nfat2* in the E μ -TCL1 mice leads to an accelerated and more aggressive development of the disease resembling those evidenced during Richter transformation.
- 2) Ablation of *Nfat2* results in the specific downregulation of several important NFAT2 target genes involved in lymphocytic anergy such as *Cbl-b*, *Grail*, *Egr2* and *Lck* and in cell cycle progression.
- 3) A differential phosphorylation status of two Src kinases, *Lyn* and *Lck* is evidenced both in the murine model and in CLL or Richter samples that seems to be correlated with the transformation.

Although the analysis based on the novel murine model is interesting, some of the conclusions should be reinforced and some concerns remain to be clarified.

Major Comments:

1. The experiments with human samples are always comparing normal B cells, CLL cells and eventually Richter transformed cells (Fig 1, 6 and 7). Taking into account the importance of NFAT2 expression for the development of CD5⁺ CD19⁺ "B1a-like" cells and its differential expression between lymphoid subpopulations, it is of importance to compare experimentally equivalent subpopulations between healthy controls, CLL samples and Richter transformations and to provide information on the purification procedures as well as phenotypic features of the populations examined before any conclusion.

Response: We agree with the reviewer that every effort should be taken to compare functionally equivalent lymphoid subpopulations in the experiments. In the revised version of the manuscript we analyzed CLL cells from patients with indolent and ag-

Universitätsklinikum Tübingen

Anstalt des öffentlichen Rechts
Sitz Tübingen
Geissweg 3 - 72076 Tübingen
Telefon (07071) 29-0
www.medizin.uni-tuebingen.de
Steuer-Nr. 86156/09402
USt.-ID: DE 146 889 674

Aufsichtsrat

Hartmut Schrade (Vorsitzender)

Vorstand

Prof. Dr. Michael Bamberg (Vorsitzender)
Gabriele Sonntag (Stellv. Vorsitzende)*
Prof. Dr. Karl Ulrich Bartz-Schmidt
Prof. Dr. Ingo B. Autenrieth
Klaus Tischler

Banken

Baden-Württembergische Bank Stuttgart
(BLZ 600 501 01) Konto-Nr. 7477 5037 93
IBAN: DE41 6005 0101 7477 5037 93
SWIFT-Nr.: SOLADEST
Kreissparkasse Tübingen
(BLZ 641 500 20) Konto-Nr. 14 144
IBAN: DE79 6415 0020 0000 0141 44

gressive forms of CLL as well as lymph node samples from patients with indolent CLL and patients with Richter's syndrome. As outlined above, this approach is somewhat complicated by the fact that CLL is usually diagnosed in the peripheral blood and no lymph node or bone marrow biopsies are performed according to clinical guidelines. Richter's syndrome on the other hand is diagnosed in lymph node biopsies or other tissue samples, while these kind of biopsies are rarely available for untransformed CLL cases (indolent and aggressive). Human CD19+ B cells and CD19+ CLL cells were isolated using the MACS-based "Untouched Human B Cell Isolation Kit" from Miltenyi Biotec as outlined in the methods section. After the isolation procedure, the purity of isolated CD19+ cell populations was assessed by flow cytometry and was routinely found to be >90%. For the analysis of cases with documented Richter's transformation, we used the respective lymph node biopsies of those patients and available lymph node biopsies from patients with indolent CLL. To provide additional evidence that LCK expression is indeed significantly downregulated in lymph nodes affected with Richter's syndrome in comparison with cases of indolent CLL, we now have included immunohistochemical stains of the respective lymph node sections in the revised version of the manuscript.

2. Flow cytometry analysis of the three murine models (Fig 2 and 3) shows as already documented an increase of the CD19+CD5+ subpopulation in E μ -TCL1 transgenic mice and further increase upon ablation of *Nfat2*. However, while, as expected, E μ -TCL1 transgenics show modification of this population only, the B-cell specific ablation of *Nfat2* results in a large decrease of other cell populations that should not be directly impacted such as the CD19-CD5+ population. An exhaustive analysis of the various lymphoid subpopulations in which NFAT2 has a relevant impact should be provided. This point is of high importance since the evaluation of the functional impact of the ablation is afterward performed as a result on the whole population and not in a particular. Of note Figures 2c and d indicate clearly that *Nfat2* ablation impacts not only CD5+CD19+ cells but also other white blood cells and similar modifications are observed in spleens in Figure 3d.

Response: We agree with the reviewer that Figures 2a shows a decrease of other cell populations (e.g. CD19-CD5+ lymphocytes) in addition to the expansion of CD19+

Universitätsklinikum Tübingen

Anstalt des öffentlichen Rechts
Sitz Tübingen
Geissweg 3 - 72076 Tübingen
Telefon (07071) 29-0
www.medizin.uni-tuebingen.de
Steuer-Nr. 86156/09402
USt.-ID: DE 146 889 674

Aufsichtsrat

Hartmut Schrade (Vorsitzender)

Vorstand

Prof. Dr. Michael Bamberg (Vorsitzender)
Gabriele Sonntag (Stellv. Vorsitzende)*
Prof. Dr. Karl Ulrich Bartz-Schmidt
Prof. Dr. Ingo B. Autenrieth
Klaus Tischler

Banken

Baden-Württembergische Bank Stuttgart
(BLZ 600 501 01) Konto-Nr. 7477 5037 93
IBAN: DE41 6005 0101 7477 5037 93
SWIFT-Nr.: SOLADEST
Kreissparkasse Tübingen
(BLZ 641 500 20) Konto-Nr. 14 144
IBAN: DE79 6415 0020 0000 0141 44

CD5+ B cells. This effect is largely due to the progressive leukemia infiltration of the bone marrow and other lymphoid organs and is significantly more pronounced in the TCL1 *Nfat2*^{-/-} cohort. As requested by the reviewer, we now provide an extensive analysis of other lymphoid subpopulations (T cells, follicular B cells, marginal zone B cells, memory B cells) which is included in Supplementary Figure 3 of the revised version of the manuscript. Of note, there was no significant difference detectable in the different experimental cohorts with respect to the lymphoid subpopulations analyzed.

3. Relationship between accelerated tumor cell turnover (notably annexinV labeling Fig 2 e and f), cell cycle progression and aggressiveness of the disease should be more documented (comparison with NSG transplanted animals? Upon transformation to aggressive disease?)

Reponse: To more extensively characterize the effects of *Nfat2* ablation on tumor cell turnover, cell cycle progression and disease aggressiveness, we performed an additional experiment in which we transplanted NSG mice with CLL cells from TCL1 *Nfat2*^{+/+} or TCL1 *Nfat2*^{-/-} animals (Supplementary Figure 5 of the revised version of the manuscript). 5 weeks after transplantation, proliferation and apoptosis were assessed by in vivo BrdU and AnnexinV staining and subsequent analysis by flow cytometry. As expected, CLL cells from TCL1 *Nfat2*^{-/-} animals exhibited a significant higher proliferation capacity in the peripheral blood as well as in the bone marrow. Apoptosis was also significantly reduced in CLL cells from TCL1 *Nfat2*^{-/-} mice in the peripheral blood.

4. The authors conclude on the absence of evidence for a relevant gene dosage effect. However in the E μ -TCL1 context, Kaplan-Meier plots in Figure 3a and supplementary Figure 1 (TCL1 *Nfat2*^{+/+}) indicate different 50% survival (325 versus 297 or 291 days). If the difference between +/+ and +/- animals is 325 versus 267 and not 291 versus 267 would this increase generate significance? Values should be verified and compiled in a single Figure.

Response: We have thoroughly verified the values for median survival of the individual cohorts and have now compiled them in a single figure which is included as Supplementary Figure 4 in the revised version of the manuscript. The *Nfat2*^{+/+} and *Nfat2*^{-/-}

Universitätsklinikum Tübingen

Anstalt des öffentlichen Rechts
Sitz Tübingen
Geissweg 3 - 72076 Tübingen
Telefon (07071) 29-0
www.medizin.uni-tuebingen.de
Steuer-Nr. 86156/09402
USt.-ID: DE 146 889 674

Aufsichtsrat

Hartmut Schrade (Vorsitzender)

Vorstand

Prof. Dr. Michael Bamberg (Vorsitzender)
Gabriele Sonntag (Stellv. Vorsitzende)*
Prof. Dr. Karl Ulrich Bartz-Schmidt
Prof. Dr. Ingo B. Autenrieth
Klaus Tischler

Banken

Baden-Württembergische Bank Stuttgart
(BLZ 600 501 01) Konto-Nr. 7477 5037 93
IBAN: DE41 6005 0101 7477 5037 93
SWIFT-Nr.: SOLADEST
Kreissparkasse Tübingen
(BLZ 641 500 20) Konto-Nr. 14 144
IBAN: DE79 6415 0020 0000 0141 44

cohorts without concomitant expression of the TCL1 transgene did not exhibit any evidence of disease during the 400 day follow up period of the experiment. All animals were alive at the end of the study. Median survival was 325 days for the TCL1 *Nfat2*^{+/+} cohort while it was 201 days for the TCL1 *Nfat2*^{-/-} cohort, which reached clear statistical significance both in the Log Rank test and in the Gehan-Breslow-Wilcoxon test. The median survival of the heterozygous TCL1 *Nfat2*^{+/-} animals was 276 days. While we agree with the reviewer that there might be a tendency towards shorter survival in this cohort, the comparison with the TCL1 *Nfat2*^{+/+} did not reach statistical significance in the Log Rank test and in the Gehan-Breslow-Wilcoxon test. The tendency towards shorter survival without reaching statistical significance is now discussed in more detail in the revised version of the manuscript.

5. Conclusions raised from the experiments in Figure 6g are unclear and should be more controlled. Characterization of a colocalization of the Src kinases Lyn and Lck with the antigen receptor is based on a reporter and amplification system. The minimal amount of proteins necessary to visualize the colocalization is not assessed and the experiment not quantitative. Since TCL1*Nfat2*^{-/-} cells express very low levels of Lck as compared to TCL1 *Nfat2*^{+/+} cells the authors should provide cell staining indicating the residual levels of expression of Lck in the cells before concluding on the absence of the colocalization of the protein with CD79a. Also, Lyn kinase recruitment to the complex seems different between the two mouse models and the antibodies used are not specific of the activated or closed forms of the proteins. Therefore it is difficult to conclude on the absence of recruitment under resting conditions as compared to BCR engagement. Staining for protein expression and activation should be provided in order to confirm the conclusions.

Response: Single antibody stainings for LCK and LYN at resting conditions have been performed as requested by the reviewer and are now included as Supplementary Figure 7 in the revised version of the manuscript. While the expression levels for LYN were comparable in the two experimental cohorts, LCK expression was virtually absent in CLL cells from the TCL1 *Nfat2*^{-/-} cohort. We also tried to detect the activated form of LYN in these cells but unfortunately were unable to do so because the antibodies were not sufficiently specific.

Universitätsklinikum Tübingen

Anstalt des öffentlichen Rechts
Sitz Tübingen
Geissweg 3 - 72076 Tübingen
Telefon (07071) 29-0
www.medizin.uni-tuebingen.de
Steuer-Nr. 86156/09402
USt.-ID: DE 146 889 674

Aufsichtsrat

Hartmut Schrade (Vorsitzender)

Vorstand

Prof. Dr. Michael Bamberg (Vorsitzender)
Gabriele Sonntag (Stellv. Vorsitzende)*
Prof. Dr. Karl Ulrich Bartz-Schmidt
Prof. Dr. Ingo B. Autenrieth
Klaus Tischler

Banken

Baden-Württembergische Bank Stuttgart
(BLZ 600 501 01) Konto-Nr. 7477 5037 93
IBAN: DE41 6005 0101 7477 5037 93
SWIFT-Nr.: SOLADEST
Kreissparkasse Tübingen
(BLZ 641 500 20) Konto-Nr. 14 144
IBAN: DE79 6415 0020 0000 0141 44

6. The implication of NFAT2 in B CLL cells anergy is documented through the analysis of the phosphorylation state of Lyn and Lck (Fig 7). The authors argue for a balance between the inhibitory phosphorylation of Lyn and an activated phosphorylation status of Lck in anergic cells. Both phosphorylation status are lowered in TCL1 *Nfat2*^{-/-} cells. However since p394-Lck is very low in TCL1 *Nfat2*^{-/-} cells one should expect that the activation is then due to the p397 Lyn active form. Due to the interplay of the two kinases and the shortage of documentation on the role of Lck in B cells the authors should clarify this important point. A quantitative analysis of the relative level of expression of Lyn and Lck should be performed altogether with the analysis of both inactive and active phosphorylation forms of the kinases in the two mice model upon BCR stimulation. p397 Lyn and p505 Lck status are clearly neglected. Similarly activation status of Lck would have been of interest in human samples in Fig 7g.

Response: As suggested by the reviewer, we have now also included an analysis of p505-Lck in both experimental cohorts in Figure 7 in the revised version of the manuscript. As stated above, we also tried to detect the activated form of LYN in the CLL cells of those cohorts but were unfortunately unable to do so due to specificity issues with the used antibodies. We have now also performed a quantitative analysis of the relative expression levels of LYN, LCK and other signaling proteins as suggested by the reviewer which is included as the new Supplementary Figure 8. This analysis basically confirms our original interpretation of the data. Total protein expression of LYN was not significantly different between the two experimental cohorts while the inactive form p507-LYN was clearly overexpressed in the anergic TCL1 *Nfat2*^{+/+} cohort. SYK expression was comparable in both cohorts according to the quantitative analysis while BCR stimulation induced SYK phosphorylation and activation in the *Nfat2*-ablated cohort but not in anergic CLL cells with intact NFAT2 expression. LCK was overexpressed and induced in CLL cells from the TCL1 *Nfat2*^{+/+} cohort but remained low in CLL cells with *Nfat2* deletion even after BCR stimulation. Expression levels of the inactive form p505-LCK were comparable in both cohorts while the activated form p394-LCK was significantly overexpressed in anergic CLL cells from TCL1 *Nfat2*^{+/+} animals. While expression levels of ERK1/2 were comparable in both cohorts, anergic CLL cells from TCL1 mice with intact NFAT2 expression showed evidence of constitutive activation of ERK1/2. In TCL1 *Nfat2*^{-/-} animals on the other hand, ERK1/2 was

Universitätsklinikum Tübingen

Anstalt des öffentlichen Rechts
Sitz Tübingen
Geissweg 3 - 72076 Tübingen
Telefon (07071) 29-0
www.medizin.uni-tuebingen.de
Steuer-Nr. 86156/09402
USt.-ID: DE 146 889 674

Aufsichtsrat

Hartmut Schrade (Vorsitzender)

Vorstand

Prof. Dr. Michael Bamberg (Vorsitzender)
Gabriele Sonntag (Stellv. Vorsitzende)*
Prof. Dr. Karl Ulrich Bartz-Schmidt
Prof. Dr. Ingo B. Autenrieth
Klaus Tischler

Banken

Baden-Württembergische Bank Stuttgart
(BLZ 600 501 01) Konto-Nr. 7477 5037 93
IBAN: DE41 6005 0101 7477 5037 93
SWIFT-Nr.: SOLADEST
Kreissparkasse Tübingen
(BLZ 641 500 20) Konto-Nr. 14 144
IBAN: DE79 6415 0020 0000 0141 44

completely inactive without BCR stimulation and changed its activation status only upon α lgM treatment.

Furthermore, we have now also analyzed human samples from patients with indolent and aggressive forms of CLL for the activation status of LCK which is now included in revised Figure 6d-f. While patients with indolent CLL exhibited significant activation of LCK, its activation status was comparable to physiological B cells in patients with aggressive CLL.

Minor Comments:

1. It would be of interest to visualize the relative level of NFAT2 protein expression in NFAT2^{+/+} mice in order to correlate their calcium mobilization with the other mice models expressing high NFAT2 (TCL1 NFAT2^{+/+}) or not (TCL1 NFAT2^{-/-}) (Fig 1g).

Response: As suggested by the reviewer, the relative expression levels of NFAT2 in the different mouse strains is now visualized within a single Western Blot which is included as Supplementary Figure 2a in the revised version of the manuscript.

2. Page 6: numbers of days should be identical to those in figures.

Response: The respective corrections have been made in the revised version of the manuscript.

3. Page 7: Fig 4, Supplementary Figure 2 should be corrected.

Response: The respective correction was made in the revised version of the manuscript.

4. Page 9: the sentence Irrelevant IgG and an anti acetyl-histone H3 antibody served as positive and negative controls, respectively should be modified.

Universitätsklinikum Tübingen

Anstalt des öffentlichen Rechts
Sitz Tübingen
Geissweg 3 - 72076 Tübingen
Telefon (07071) 29-0
www.medizin.uni-tuebingen.de
Steuer-Nr. 86156/09402
USt.-ID: DE 146 889 674

Aufsichtsrat

Hartmut Schrade (Vorsitzender)

Vorstand

Prof. Dr. Michael Bamberg (Vorsitzender)
Gabriele Sonntag (Stellv. Vorsitzende)*
Prof. Dr. Karl Ulrich Bartz-Schmidt
Prof. Dr. Ingo B. Autenrieth
Klaus Tischler

Banken

Baden-Württembergische Bank Stuttgart
(BLZ 600 501 01) Konto-Nr. 7477 5037 93
IBAN: DE41 6005 0101 7477 5037 93
SWIFT-Nr.: SOLADEST
Kreissparkasse Tübingen
(BLZ 641 500 20) Konto-Nr. 14 144
IBAN: DE79 6415 0020 0000 0141 44

Response: The respective adjustments were made in the revised versions of the manuscript.

Reviewer #3:

Remarks to the author:

In this manuscript Marklin et al investigate the role and significance of the transcription factor NFAT2 in chronic lymphocytic leukemia (CLL) starting from the well known notion that the constitutive activation of Nfat2 is a hallmark of the anergic phenotype in CLL. They essentially use the Tcl1 CLL mouse model but also present data (albeit less numerous and impressive) on human samples. Their main result is that the ablation of Nfat2 in Tcl1 mice leads to the loss of the anergic CLL phenotype and favours the transformation of an indolent disease into an aggressive one which is reminiscent of the Richter transformation in human CLL. In other words the Authors molecularly characterize the role of a transcription factor which was already known to pinpoint the anergic phenotype of CLL, but whose molecular definition was not known, certainly not to the detailed degree shown by the Authors. Furthermore the Authors find a gene expression signature of anergic CLL cells which consists of several NFAT2-dependant genes and may have a number of implications. The take home message is that NFAT2 is a crucial regulator of the anergic phenotype in CLL whose loss causes the transition from an indolent to an aggressive disease. These findings are of interest but raise a number of questions.

Major Comments:

1. The first question is: what regulates the regulator in CLL? To be more specific why NFAT2 at a certain point of the natural history of CLL is or becomes inactivated and what leads to its inactivation in human beings, considering that this inactivation may lead to the aggressive transformation ?

Universitätsklinikum Tübingen

Anstalt des öffentlichen Rechts
Sitz Tübingen
Geissweg 3 - 72076 Tübingen
Telefon (07071) 29-0
www.medizin.uni-tuebingen.de
Steuer-Nr. 86156/09402
USt.-ID: DE 146 889 674

Aufsichtsrat

Hartmut Schrade (Vorsitzender)

Vorstand

Prof. Dr. Michael Bamberg (Vorsitzender)
Gabriele Sonntag (Stellv. Vorsitzende)*
Prof. Dr. Karl Ulrich Bartz-Schmidt
Prof. Dr. Ingo B. Autenrieth
Klaus Tischler

Banken

Baden-Württembergische Bank Stuttgart
(BLZ 600 501 01) Konto-Nr. 7477 5037 93
IBAN: DE41 6005 0101 7477 5037 93
SWIFT-Nr.: SOLADEST
Kreissparkasse Tübingen
(BLZ 641 500 20) Konto-Nr. 14 144
IBAN: DE79 6415 0020 0000 0141 44

Response: We agree with the reviewer that this is a very important question resulting from the results of our study. Sequencing of the cancer genome from patients with aggressive/accelerated forms of CLL or Richter's syndrome have not revealed any evidence for inactivating mutations in NFAT2 as a potential explanation for this phenomenon. Multiple lines of evidence point to epigenetic changes in the NFAT2 promoter region which are acquired during disease evolution/acceleration and result in a downregulation of NFAT2 expression in the tumor cells of patients with accelerated or transformed CLL. We are currently addressing this question in a bigger cohort of human patients and hope to be able report more precise results in the future. Furthermore, this important question is now thoroughly discussed in the revised version of the manuscript.

2. A second major question is related to the fact that a number of genes have already been found or suggested to be involved in the transformation of CLL into Richter's syndrome. How do these genes relate to NFAT2? Is NFAT2 sort of a hub? Are the already described genes capable of influencing or downregulating NFAT2 so that its role of "anergic brake" is lost?

Response: Downregulation or loss of CDKN2A and disruption of TP53 have both been described to be involved in the pathogenesis of Richter's syndrome. We have therefore assessed the expression levels of the two genes in our two experimental cohorts and found a statistically significant downregulation of CDKN2A and TP53 expression in CLL cells from TCL1 *Nfat2*^{-/-} mice (Supplementary Figure 9). Detailed studies on how these genes can interact with NFAT2 and influence its expression have not been undertaken but would be of great interest for a better understanding of the pathophysiology of this disease.

3. Though admittedly it is not easy to have at hand many cases of Richter's syndrome, the question is whether the number of human samples is sufficient to reproduce in humans the mouse data. Also what happens to NFAT2 in the cells of patients who are treatment-unresponsive, aggressive but are NOT in histologic transformation? These patients are numerous. These questions should be at least asked otherwise the discussion remains a mere repetition of the results.

Universitätsklinikum Tübingen

Anstalt des öffentlichen Rechts
Sitz Tübingen
Geissweg 3 - 72076 Tübingen
Telefon (07071) 29-0
www.medizin.uni-tuebingen.de
Steuer-Nr. 86156/09402
USt.-ID: DE 146 889 674

Aufsichtsrat

Hartmut Schrade (Vorsitzender)

Vorstand

Prof. Dr. Michael Bamberg (Vorsitzender)
Gabriele Sonntag (Stellv. Vorsitzende)*
Prof. Dr. Karl Ulrich Bartz-Schmidt
Prof. Dr. Ingo B. Autenrieth
Klaus Tischler

Banken

Baden-Württembergische Bank Stuttgart
(BLZ 600 501 01) Konto-Nr. 7477 5037 93
IBAN: DE41 6005 0101 7477 5037 93
SWIFT-Nr.: SOLADEST
Kreissparkasse Tübingen
(BLZ 641 500 20) Konto-Nr. 14 144
IBAN: DE79 6415 0020 0000 0141 44

Response: We agree with the reviewer that our cohort of human patient sample is too small to draw general conclusions. The purpose of including human samples in this study was to show that observations in line with our newly developed mouse model can also be made in human CLL patients. These limitations are now acknowledged more precisely in the revised version of the manuscript. Also, we have included data from patients with aggressive CLL in the revised version of the manuscript (Figures 1, 6 and 7). In summary, our data demonstrate that NFAT2 and LCK overexpression and activation is prevalent in indolent cases of CLL and correlates with the anergic phenotype of the disease.

4. On a more literary ground I find the Introduction and especially the Discussion too long and verbose.

Response: We have carefully edited the Introduction and the Discussion sections of the revised version of the manuscript to make them more concise.

Please let us know if there should be any additional questions.

Sincerely,

Martin R. Müller, MD, PhD
University of Tübingen
Dept. of Oncology, Hematology and Immunology
Otfried-Müller-Str. 10
72076 Tübingen
Germany

Universitätsklinikum Tübingen

Anstalt des öffentlichen Rechts
Sitz Tübingen
Geissweg 3 - 72076 Tübingen
Telefon (07071) 29-0
www.medizin.uni-tuebingen.de
Steuer-Nr. 86156/09402
USt.-ID: DE 146 889 674

Aufsichtsrat

Hartmut Schrade (Vorsitzender)

Vorstand

Prof. Dr. Michael Bamberg (Vorsitzender)
Gabriele Sonntag (Stellv. Vorsitzende)*
Prof. Dr. Karl Ulrich Bartz-Schmidt
Prof. Dr. Ingo B. Autenrieth
Klaus Tischler

Banken

Baden-Württembergische Bank Stuttgart
(BLZ 600 501 01) Konto-Nr. 7477 5037 93
IBAN: DE41 6005 0101 7477 5037 93
SWIFT-Nr.: SOLADEST
Kreissparkasse Tübingen
(BLZ 641 500 20) Konto-Nr. 14 144
IBAN: DE79 6415 0020 0000 0141 44

Eberhard-Karls-Universität
UKT
Universitätsklinikum Tübingen

Medizinische Klinik, Otfried-Müller-Str. 10, D-72076 Tübingen

**Universitätsklinikum Tübingen
Medizinische Klinik und Poliklinik**

Abteilung und Lehrstuhl II

*Onkologie
Hämatologie
Immunologie
Rheumatologie
Pulmologie*

*Ärztlicher Direktor
Prof. Dr. med. Lothar Kanz*

*Otfried-Müller-Straße 10
D-72076 Tübingen
Telefon: +49/7071/2 98 27 26
Telefax: +49/7071/29 36 71
E-mail: lothar.kanz@med.uni-tuebingen.de
www.onkologie-tuebingen.de*

Martin R. Müller, MD, PhD

July 20, 2017

Response to reviewers' comments:

Reviewer #2:

Remarks to the author:

The reviewer acknowledges the revised version of the manuscript by Märklin et al. The authors designed important and informative new experiments in order to answer the numerous comments addressed by the reviewers. The experiments confirmed important findings on the TCL1Nfat2^{-/-} mice notably that NFAT2 ablation leads to an earlier and more aggressive development of the disease resembling those evidenced during Richter transformation. It also emphasizes the role of NFAT2 in the regulation of anergy-linked genes in line with a similar regulation observed in CLL patients at different stages. However, several responses are still elusive and do not address the comments previously raised.

Universitätsklinikum Tübingen

Anstalt des öffentlichen Rechts
Sitz Tübingen
Geissweg 3 - 72076 Tübingen
Telefon (07071) 29-0
www.medicin.uni-tuebingen.de
Steuer-Nr. 86156/09402
USt.-ID: DE 146 889 674

Aufsichtsrat

Hartmut Schrade (Vorsitzender)
Vorstand
Prof. Dr. Michael Bamberg (Vorsitzender)
Gabriele Sonntag (Stellv. Vorsitzende)*
Prof. Dr. Karl Ulrich Bartz-Schmidt
Prof. Dr. Ingo B. Autenrieth
Klaus Tischler

Banken

Baden-Württembergische Bank Stuttgart
(BLZ 600 501 01) Konto-Nr. 7477 5037 93
IBAN: DE41 6005 0101 7477 5037 93
SWIFT-Nr.: SOLADEST
Kreissparkasse Tübingen
(BLZ 641 500 20) Konto-Nr. 14 144
IBAN: DE79 6415 0020 0000 0141 44

Major Comments:

1. The confusion raised in the initial manuscript between the anergic phenotype of several CLL cells and the indolent clinical outcome with only a focus on NFAT and Lck expression has been addressed during edition of the manuscript but is still only very limited on an experimental point of view. It is important that the authors provide direct evidences of the activation status of the cells for patients with “indolent” or “aggressive” outcome (Calcium or phosphorylation of effectors and not only of Lck). Moreover, the papers indicated in reference to this notion do not conclude precisely on this specific link between anergy and indolent disease (Page 4 introduction, p13 Discussion ref 9, 11, 23, 24, 29, 30). This point should be appropriately addressed and clarified in CLL patients with different profiles used in the study.

Response: We agree with reviewer#2 that anergic phenotype does not necessarily mean indolent clinical outcome. To prevent any kind of confusion in this respect we have already made the necessary adjustments in the last version of the manuscript. While a systematic analysis of this question was beyond the scope of this study, we have typically observed an anergic phenotype in the CLL cells from patients with indolent disease and enhanced/prolonged calcium flux and BCR signaling in CLL cells from patients with aggressive disease. To underscore this observation and to provide direct evidence of the activation status we have now included calcium mobilization assays of CLL cells from patients with indolent (n=2) and aggressive disease (n=2) as the new Supplementary Figure 9 in the revised version of the manuscript. We agree with reviewer#2 that the references provided in the manuscript do not conclude precisely on the specific link between anergy and indolent disease and have modified the manuscript accordingly.

2. There is still some controversial explanation between the flow cytometry data in figure 2 and 3 and explanation in the text on page 6 (increase of the CD19+ CD5- during later disease stages with internalization of CD5 is not seen in figure 2). Additionally, explanation on pronounced tumor infiltrate is not shown and sufficient to

Universitätsklinikum Tübingen

Anstalt des öffentlichen Rechts
Sitz Tübingen
Geissweg 3 - 72076 Tübingen
Telefon (07071) 29-0
www.medizin.uni-tuebingen.de
Steuer-Nr. 86156/09402
USt-ID: DE 146 889 674

Aufsichtsrat

Hartmut Schrade (Vorsitzender)
Vorstand
Prof. Dr. Michael Bamberg (Vorsitzender)
Gabriele Sonntag (Stellv. Vorsitzende)*
Prof. Dr. Karl Ulrich Bartz-Schmidt
Prof. Dr. Ingo B. Autenrieth
Klaus Tischler

Banken

Baden-Württembergische Bank Stuttgart
(BLZ 600 501 01) Konto-Nr. 7477 5037 93
IBAN: DE41 6005 0101 7477 5037 93
SWIFT-Nr.: SOLADEST
Kreissparkasse Tübingen
(BLZ 641 500 20) Konto-Nr. 14 144
IBAN: DE79 6415 0020 0000 0141 44

explain the dot plots in figures 2 and 3 for CD19⁻ CD5⁺ or in figure 4 while sup figure 3 shows no alteration by NFAT ablation for T cell population (CD3⁺). Regarding the important role of NFAT2 in various lymphoid lineages these findings should be more precisely addressed and clarified in the manuscript (pages 6 and 8).

Response: As shown in Supplementary Figure 2, CD5 expression is progressively downregulated on CLL cells during later stages of the disease. This at least in part explains the observed increase of the CD5-CD19⁺ population. We do not agree with the reviewer that this effect cannot be detected in Figure 2a. The neoplastic CD19⁺ cell population from TCL1 Nfat2^{-/-} mice does exhibit a clear shift to the left indicative of CD5 downregulation. In Figure 2a, the peripheral blood of mice at an age of 28 weeks was analyzed by flow cytometry. At this time point, all TCL1⁺ mice have developed overt disease and extensive bone marrow infiltration. This is particularly more pronounced in the TCL1 Nfat2^{-/-} cohort and in our opinion can very well explain the progressive disappearance of other cell populations in the blood of these animals. It is important to keep in mind that our analysis of other lymphoid subpopulations was undertaken in mice at an age of 20 weeks when they just started to develop leukemic disease. It is therefore not surprising that they do not show any significant alterations of the the CD3⁺ T cell population.

3. Explanation on the recruitment of Lyn versus Lck to the CD79a is quite misleading since in TCL1 Nfat2^{-/-} resting cells both Lck and Lyn are downregulated (sup Figure 7) and Lyn is not recruited (resting cells) anymore or weakly detected (upon stimulation) (Figure 6). Since technical problems do not allow detection of the activated form of Lyn but Y507 P-Lyn is downregulated in TCL1 Nfat2^{-/-} mice as compared to TCL1 Nfat2^{+/+} (Figure 7d) it would be important to provide additional comments using either P394 Lck or P Lyn507 in Figure 6h or at least use the similar antibodies for patient samples in figure 7g before concluding on the replacement of one protein by the other.

Response: As shown in Supplementary Figure 7, LYN is expressed at comparative levels in TCL1 Nfat2^{+/+} and TCL1 Nfat2^{-/-} mice, while LCK expression is clear-

Universitätsklinikum Tübingen

Anstalt des öffentlichen Rechts
Sitz Tübingen
Geissweg 3 - 72076 Tübingen
Telefon (07071) 29-0
www.medizin.uni-tuebingen.de
Steuer-Nr. 86156/09402
USt-ID: DE 146 889 674

Aufsichtsrat

Hartmut Schrade (Vorsitzender)
Vorstand
Prof. Dr. Michael Bamberg (Vorsitzender)
Gabriele Sonntag (Stellv. Vorsitzende)*
Prof. Dr. Karl Ulrich Bartz-Schmidt
Prof. Dr. Ingo B. Autenrieth
Klaus Tischler

Banken

Baden-Württembergische Bank Stuttgart
(BLZ 600 501 01) Konto-Nr. 7477 5037 93
IBAN: DE41 6005 0101 7477 5037 93
SWIFT-Nr.: SOLADEST
Kreissparkasse Tübingen
(BLZ 641 500 20) Konto-Nr. 14 144
IBAN: DE79 6415 0020 0000 0141 44

ly downregulated in TCL1 Nfat2^{-/-} mice. It is important here to take into account that there are obviously less cells on the slide of the TCL1 Nfat2^{-/-} cohort. The purpose of the experiment displayed in Fig. 6h was to demonstrate that in CLL cells with intact NFAT2 expression, LCK is expressed and does colocalize with the BCR complex upon IgM stimulation. In CLL cells with NFAT2 deletion on the other hand, LCK was completely absent from the BCR, both at resting conditions and after BCR stimulation. In summary, these data show that LCK is a target gene of the transcription factor NFAT2 in CLL cells and colocalizes with CD79a upon BCR engagement. These are the sole conclusions we are drawing from this experiment.

Universitätsklinikum Tübingen

Anstalt des öffentlichen Rechts
Sitz Tübingen
Geissweg 3 - 72076 Tübingen
Telefon (07071) 29-0
www.medizin.uni-tuebingen.de
Steuer-Nr. 86156/09402
USt-ID: DE 146 889 674

Aufsichtsrat

Hartmut Schrade (Vorsitzender)

Vorstand

Prof. Dr. Michael Bamberg (Vorsitzender)
Gabriele Sonntag (Stellv. Vorsitzende)*
Prof. Dr. Karl Ulrich Bartz-Schmidt
Prof. Dr. Ingo B. Autenrieth
Klaus Tischler

Banken

Baden-Württembergische Bank Stuttgart
(BLZ 600 501 01) Konto-Nr. 7477 5037 93
IBAN: DE41 6005 0101 7477 5037 93
SWIFT-Nr.: SOLADEST
Kreissparkasse Tübingen
(BLZ 641 500 20) Konto-Nr. 14 144
IBAN: DE79 6415 0020 0000 0141 44